medicinal chemistry/organic chemistry/synthetic chemistry

anti-cancer agents, biomolecules, carcinoma cell lines, cytotoxicity, pyrrolidine-2-carboxamides

**Authors for correspondence:**
Xiaoguang Bao
e-mail: xgbao@suda.edu.cn
Oluwole Familoni
e-mail: familonio@unilag.edu.ng

This article has been edited by the Royal Society of Chemistry, including the commissioning, peer review process and editorial aspects up to the point of acceptance.

# Synthesis and *in vitro* anticancer activities of substituted *N*-(4′-nitrophenyl)-L-prolinamides

Adejoke Osinubi[1,2,4], Josephat Izunobi[1],
Xiaoguang Bao[2], Olayinka Asekun[1], Jiehong Kong[3],
Chunshan Gui[3] and Oluwole Familoni[1]

[1]Department of Chemistry, University of Lagos, Akoka-Yaba, Lagos, Nigeria
[2]College of Chemistry, Chemical Engineering and Material Science, and [3]College of Pharmaceutical Sciences, Soochow University Suzhou, Jiangsu 215123, People's Republic of China
[4]Chemical Sciences Department, College of Science and Information Technology, Tai Solarin University of Education, P.M.B. 2118, Ijebu Ode, Ogun Postal, Nigeria

AO, 0000-0002-6719-305X; JI, 0000-0002-6315-4969; XB, 0000-0001-7190-8866; OA, 0000-0002-3875-003X; JK, 0000-0002-7772-4395; CG, 0000-0001-9296-0911; OF, 0000-0002-5480-232X

Prolinamides are present in secondary metabolites and have wide-ranging biological properties as well as antimicrobial and cytotoxic activities. *N*-(4′-substituted phenyl)-L-prolinamides **4a–4w** were synthesized in two steps, starting from the condensation of *p*-fluoronitrobenzene **1a–1b** with L-proline **2a–2b**, under aqueous–alcoholic basic conditions to afford *N*-aryl-L-prolines **3a–3c**, which underwent amidation via a two-stage, one-pot reaction involving $SOCl_2$ and amines, to furnish L-prolinamides in 20–80% yield. The cytotoxicities of **4a–4w** against four human carcinoma cell lines (SGC7901, HCT-116, HepG2 and A549) were evaluated by 3-(4,5-dimethylthiazol-2-yl)-2,5-diphenyltetrazolium bromide (MTT) assay; with good tumour inhibitory activities ($79.50 \pm 1.24\%$–$50.04 \pm 1.45\%$) against HepG2. **4a** exhibited the best anti-tumour activity against A549 with percentage cell inhibition of $95.41 \pm 0.67\%$ at 100 μM. Likewise, **4s** ($70.13 \pm 3.41\%$) and **4u** ($83.36 \pm 1.70\%$) displayed stronger antineoplastic potencies against A549 than the standard, 5-fluorouracil ($64.29 \pm 2.09\%$), whereas **4a** ($93.33 \pm 1.36\%$) and **4u** ($81.29 \pm 2.32\%$) outperformed the reference ($81.20 \pm 0.08\%$) against HCT-116. SGC7901 showed lower percentage cell viabilities with **4u** ($8.02 \pm 1.54\%$) and **4w**

ROYAL SOCIETY OF CHEMISTRY

(27.27 ± 2.38%). These results underscore the antiproliferative efficacies of L-prolinamides while exposing **4a** and **4u** as promising broad-spectrum anti-cancer agents. Structure-activity relationship studies are discussed.

# 1. Introduction

Cancer is the uncontrolled growth and spread of cells. It is a non-communicable disease but remains the second leading cause of mortality worldwide, with over 9.6 million estimated deaths and 18.1 million new cases annually [1,2]. The identification and syntheses of chemical anti-cancer agents with wide-ranging inhibitory activities against tumour cells have continued to stimulate research across disciplines [3–5]. Though several therapy protocols have been applied in the treatment of the different types of cancer, their non-selectivity towards target tumour cells and cytotoxicity to normal cells as well as multidrug resistance, non-bioavailability and severe side effects are major limitations [6].

Peptides are typically accessed in the development of new classes of therapeutic agents because of their high activity, low immunogenicity, good biocompatibility and amenability to synthetic modifications [4] but naturally occurring peptides are often unsuitable therapeutics because of such limitations as poor chemical and physical stabilities, as well as short circulating plasma half-lives. The application of synthetic peptides has also been severely restricted by poor membrane permeability, high clearance, low systemic stability and negligible activity when administered orally [7]. However, various strategies, such as the inclusion of non-natural amino acids and their conjugation with sugars, lipids and proteins as well as the use of polyvalent peptide synthesis, have been adopted to overcome these drawbacks [8].

The *N*-functionalization of amino acids with aryl compounds has great use in organic syntheses as well as drug discovery and pharmaceutics [9,10]. *N*-aryl amino acids are, therefore, employed as inexpensive chiral building blocks and are crucial motifs in many systems of physiological importance [11]. *N*-arylated amino acid derivatives can also be incorporated into peptides and proteins to propel the development of new methods for the study of protein structures and functions, among others [7]. Pertinently, *N*-arylation reactions have been exploited in the introduction of diversity into bioactive molecules and synthesis of anti-cancer agents with improved potencies [10].

Prolinamides are carboxamides of the cyclic amino acid, proline and, therefore, possess the electronic and spatial properties of the amide bonds inherent in peptides as well as their bent secondary structures. Prolinamides find uses as reverse-turn mimetics [12], ligands and asymmetric organocatalysts [13]. Furthermore, the prolinamide moiety abounds in several molecules of biological and therapeutic importance [7], and some of these compounds have been reported to possess inhibitory activities against tumour cells [14–16] (figure 1).

Prolinamide derivatives can preferentially target i-motifs and G-quadruplexes, which are dynamic nucleic acid secondary structures, believed to play key roles in gene expression [18]. The selective recognition of the *c-MYC* G-quadruplex DNA by compounds with the prolinamide motif has been recently demonstrated [17]. The *c-MYC* oncogene is associated with cell growth, proliferation and a range of malignant tumours. Therefore, the use of prolinamides to target the promoter region of oncogenes, such as *c-MYC* and *BCL-2*, is a promising route towards developing effective cancer therapies. Additionally, the *N*-arylation of prolinamides, if synergistic, should bode well for the resulting substrates' antineoplastic potencies.

Presented herein, are the synthesis and characterization of a series of *N*-(4′-nitrophenyl)-L-prolinamides via a simple two-step reaction, starting from easily available and inexpensive *p*-halogenonitrobenzenes and L-proline. Impelled by the literature precedents [7,14–18] of prolinamide-containing anti-tumour agents; dating back to actinomycin [19], four of the most commonly occurring types of human cancers [20] (colon, liver, lung and gastric) were also selected for *in vitro* assays. Consequently, the anti-cancer properties of the synthesized L-prolinamides were evaluated against human colon (HCT-116), liver (HepG2), lung (A549) and gastric (SGC7901) carcinoma cell lines.

# 2. Results and discussion

## 2.1. Chemistry

The requisite *N*-(4′-nitrophenyl)-L-prolinamides **4a–w** were accessed via a two-step reaction as shown in scheme 1. The first step involved a base-catalysed condensation reaction, using potassium carbonate in refluxing ethanol–water (1 : 1) solution, as earlier reported [21,22], to furnish *N*-(4′-nitrophenyl)-L-prolines **3a–c** in 70–90% yield. It was found that refluxing the reaction in ethanol–water (1 : 1) solution

**Figure 1.** Some prolinamide-based anti-cancer compounds (Abarelix [7], Phosmidosine [14] and Pro-4 [17]).

**Scheme 1.** Synthetic route to N-(4'-nitrophenyl)-L-prolinamides.

was marginally better than refluxing in ethanol alone. *p*-Fluoronitrobenzene **1a** was the halogenobenzene of choice because fluorine is a good leaving group and typically affords better-yielding products in reactions as those that proceed via a nucleophilic aromatic substitution ($S_NAr$) mechanism [23,24].

The amidation of the N-(4'-nitrophenyl)-L-proline adducts **3a–3c** with appropriate amines produced the target N-(4'-nitrophenyl)-L-prolinamides **4a–4w** in a two-stage, one-pot reaction [25,26]. Different amines were employed in the amidation reaction with a view to obtaining potent anti-tumour agents with broad-spectrum activities and evaluating structure–activity relationships on their cytotoxicities. These included using primary, secondary, linear or branched amines as well as sulfonamides (table 1).

The first stage of the one-pot amidation reaction involved the activation of the acid adducts **3a–3c**, into N-(4'-nitrophenyl)pyrrolidine-2-carboxylic acid chlorides **5**, via a modified procedure [25], with thionyl chloride ($SOCl_2$). It is noteworthy that the complete removal of excess $SOCl_2$ *in situ* from the reaction medium (without exposure to moisture) was crucial to the success of the chlorination reaction. We also found that using stoichiometric equivalents of $SOCl_2$ resulted in incomplete reactions. The addition of N,N-dimethylformamide (one drop) appeared to have a catalytic effect on the formation of the acid chlorides; improving yield and reaction time.

It is noteworthy that all the synthesized L-prolinamides are solid, with melting points ranging from 74–75°C (**4a**) to 252–253°C (**4e**); with the exception of **4l**, which is a viscous oil. The prevalence of the L-prolinamides in the solid-state alludes to the presence of intra- and intermolecular hydrogen bonding and $\pi$–$\pi$-stacking.

Considering the infrared (IR) spectra of the acid adducts **3a–3c** and prolinamides **4a–4w**, the disappearance of the broad acid O–H stretching bands, observed between 3400 and 3200 cm$^{-1}$ in the IR spectra of **3a–3c**, and the downward shift of the carbonyl (C=O) stretching frequencies, from 1734–1714 cm$^{-1}$ in **3a–3c**, to 1687–1637 cm$^{-1}$ in the IR spectra of **4a–4w** were both indicative of a successful transformation. As was the appearance of the N–H stretching bands between 3318 cm$^{-1}$ and 3248 cm$^{-1}$ in the IR spectra of the secondary amides. The $^1$H- and $^{13}$C-nuclear magnetic resonance (NMR) spectroscopic data as well as mass spectrometry analysis are also in agreement with the assigned structures. Notably, the presence of rotamers (1 : 1) was evident in the $^1$H- and $^{13}$C-NMR spectra of **4a**, **4e**, **4j**, **4o** and **4w**. The target N-(4'-nitrophenyl)-L-prolinamides **4a–4w** were obtained in 20–80% yield, after purification by column chromatography on silica gel, with *n*-hexane and ethyl acetate (2 : 1) as mobile phase.

**Table 1.** Synthesis of substituted *N*-(4′-nitrophenyl)-L-prolinamides (**4a–w**).

| entry | substituents | | | | L-prolinamide | m.pt. (°C) | yield (percentage) |
|---|---|---|---|---|---|---|---|
| | R₁ | R₂ | R₃ | R₄ | | | |
| 1. | NO₂ | H | *n*-butyl | *n*-butyl | **4a** | 74–75 | 60 |
| 2. | NO₂ | H | H | *n*-butyl | **4b** | 145–147 | 48 |
| 3. | NO₂ | H | H | propyl | **4c** | 179–180 | 49 |
| 4. | NO₂ | H | H | 1-prop-1-ynyl | **4d** | 158–159 | 54 |
| 5. | NO₂ | H | H | cyclohexyl | **4e** | 252–253 | 43 |
| 6. | NO₂ | H | H | PhSO₂ | **4f** | 189–191 | 58 |
| 7. | NO₂ | H | H | *t*-butyl | **4g** | 141–142 | 72 |
| 8. | NO₂ | H | H | *p*-MePhSO₂ | **4h** | 93–95 | 51 |
| 9. | NO₂ | H | 4-morpholinyl | | **4i** | 214–216 | 45 |
| 10. | NO₂ | H | isopropyl | isopropyl | **4j** | 159–160 | 28 |
| 11. | NO₂ | H | H | *p*-MePh | **4k** | 218–220 | 80 |
| 12. | CN | H | H | *p*-MePh | **4l** | (oil) | 21 |
| 13. | NO₂ | OH | H | *p*-MePh | **4m** | 236–238 | 21 |
| 14 | NO₂ | H | 2-isoindolinyl | | **4n** | 215–217 | 45 |
| 15. | NO₂ | H | propyl | propyl | **4o** | 136–138 | 48 |

(*Continued.*)

| entry | substituents | | | | L-prolinamide | m.pt. | yield |
| --- | --- | --- | --- | --- | --- | --- | --- |
| | $R_1$ | $R_2$ | $R_3$ | $R_4$ | | (°C) | (percentage) |
| 16. | $NO_2$ | H | H | 2-pyridinyl | **4p** | 180–182 | 46 |
| 17. | $NO_2$ | H | H | benzyl | **4q** | 178–179 | 55 |
| 18. | $NO_2$ | H | H | o-PhCN | **4r** | 211–213 | 35 |
| 19. | $NO_2$ | H | methyl | phenyl | **4s** | 186–188 | 33 |
| 20. | $NO_2$ | H | H | o-PhCH$_2$OH | **4t** | 179–180 | 53 |
| 21. | $NO_2$ | H | H | p-PhNO$_2$ | **4u** | 137–139 | 44 |
| 22. | $NO_2$ | H | H | isopropyl | **4v** | 231–232 | 50 |
| 23. | $NO_2$ | H | cyclohexyl | cyclohexyl | **4w** | 140–142 | 30 |

## 2.2. Biology

The cytotoxicity analyses of L-prolinamides **4a–4w** were conducted *in vitro* with four different human cancer cell lines: gastric carcinoma (SGC7901), colon carcinoma (HCT-116), liver carcinoma (HepG2) and lung carcinoma (A549), using the 3-(4,5-dimethylthiazol-2-yl)-2,5-diphenyl tetrazolium bromide (MTT) assay [27,28]. The antiproliferative efficacy data were obtained from the cell viability assay calculations of the four carcinoma cell lines after their treatment with the L-prolinamide substrates. Cellular viability assays are generally used to quantify the number of healthy cells in a sample, while cell viability represents the number of healthy cells present in a given population [29]. The percentage cell viability, on the other hand, is the ratio of the healthy cells to the total cell population expressed as a percentage. Herein, three different prolinamide concentrations (1 µM, 10 µM and 100 µM) were incubated with each of the cell lines (cf. table 2) and 5-fluorouracil, which is an antimetabolite with antineoplastic activities used to treat multiple solid tumours, was employed as the positive control.

Table 2 highlights some interesting observations in the percentage cell viability data of L-prolinamides **4a–4w**, with each compound exhibiting inhibitory effects against one or more of the human cancer cell lines assayed. Most of the compounds examined showed that the percentage cell viability of the carcinoma cell lines decreased with increase in prolinamide concentration except **4f** against SGC7901 and HepG2, and **4h** against HepG2 and A549. Similar suspected hormetic responses [30] were detected for **4a**, **4b**, **4m** and **4r** against SGC7901 as well as **4j** and **4t** (against HepG2) and **4k** and **4v** (against A549). In addition, cell proliferation [29] appears to have occurred in some of the cell lines,

**Table 2.** Percentage cell viability (MTT assay) of L-prolinamides against four human carcinoma cell lines. (SGC7901, human gastric carcinoma cell line; HCT-116, human colon carcinoma cell line; HepG2, human liver carcinoma cell line; A549, human lung carcinoma cell line; 5-FU, 5-fluorouracil (positive control); n.d., not determined.)

| | L-prolinamide | carcinoma cell lines | | | | | | | | | | | |
| --- | --- | --- | --- | --- | --- | --- | --- | --- | --- | --- | --- | --- | --- |
| | | gastric | | | colon | | | liver | | | lung | | |
| | | SGC7901 | | | HCT-116 | | | HepG2 | | | A549 | | |
| | | 1 μM | 10 μM (%) | 100 μM | 1 μM | 10 μM (%) | 100 μM | 1 μM | 10 μM (%) | 100 μM | 1 μM | 10 μM (%) | 100 μM |
| 1. | 4a | 91.98 ± 9.12 | 87.15 ± 5.50 | 95.35 ± 2.26 | 95.11 ± 4.04 | 86.87 ± 5.90 | 6.67 ± 1.36 | 95.41 ± 0.71 | 96.49 ± 1.27 | 24.95 ± 4.09 | 95.07 ± 2.65 | 93.35 ± 1.30 | 4.59 ± 0.67 |
| 2. | 4b | 111.41 ± 20.13 | 111.98 ± 21.71 | 115.12 ± 17.74 | 93.21 ± 6.34 | 87.534.06 | 61.17 ± 4.16 | 95.46 ± 4.09 | 88.58 ± 3.60 | 77.89 ± 6.34 | 90.41 ± 1.90 | 85.23 ± 2.89 | 81.52 ± 4.12 |
| 3. | 4c | 121.46 ± 12.83 | 112.23 ± 11.01 | 97.08 ± 9.08 | 95.90 ± 6.37 | 95.80 ± 4.64 | 91.37 ± 0.95 | 88.19 ± 1.65 | 89.42 ± 2.54 | 87.36 ± 3.72 | 63.01 ± 6.75 | 51.84 ± 4.33 | 52.11 ± 3.28 |
| 4. | 4d | 109.60 ± 5.81 | 101.99 ± 17.20 | 100.48 ± 15.88 | 96.58 ± 7.44 | 86.47 ± 5.47 | 81.47 ± 5.75 | 94.10 ± 0.98 | 91.10 ± 2.57 | 94.44 ± 3.38 | 85.14 ± 2.29 | 77.15 ± 5.95 | 64.43 ± 5.47 |
| 5. | 4e | 86.70 ± 5.20 | 84.72 ± 4.16 | 87.10 ± 9.56 | 88.42 ± 7.89 | 81.12 ± 5.51 | 67.79 ± 4.68 | 88.95 ± 3.60 | 77.02 ± 6.91 | 20.50 ± 1.24 | 54.11 ± 4.96 | 52.85 ± 4.13 | 42.22 ± 4.65 |
| 6. | 4f | 87.01 ± 6.85 | 88.25 ± 6.02 | 94.48 ± 13.60 | 94.49 ± 6.77 | 96.95 ± 3.69 | 86.99 ± 6.33 | 87.53 ± 1.04 | 92.92 ± 4.76 | 102.19 ± 3.43 | 122.08 ± 8.43 | 110.09 ± 3.43 | 102.53 ± 2.20 |
| 7. | 4g | 89.41 ± 2.56 | 93.51 ± 3.62 | 90.72 ± 14.06 | 91.96 ± 9.11 | 95.97 ± 6.02 | 76.76 ± 7.02 | 89.00 ± 1.32 | 89.22 ± 1.29 | 84.12 ± 0.90 | 95.68 ± 8.49 | 95.24 ± 6.32 | 84.85 ± 4.19 |
| 8. | 4h | 95.71 ± 4.25 | 86.12 ± 463 | 71.05 ± 0.61 | 95.63 ± 637 | 104.93 ± 464 | 96.64 ± 0.95 | 79.90 ± 2.26 | 89.20 ± 2.05 | 94.03 ± 2.77 | 70.15 ± 9.28 | 72.09 ± 3.93 | 91.46 ± 4.20 |
| 9. | 4i | 90.08 ± 17.70 | 89.37 ± 13.72 | 77.68 ± 4.87 | 96.39 ± 639 | 92.24 ± 6.15 | 93.13 ± 7.95 | 82.36 ± 5.24 | 87.61 ± 4.24 | 85.84 ± 1.10 | 114.10 ± 6.56 | 97.28 ± 10.58 | 79.81 ± 2.95 |
| 10. | 4j | 81.51 ± 2.90 | 83.58 ± 1.95 | 68.41 ± 4.88 | 101.49 ± 3.59 | 101.04 ± 4.66 | 108.82 ± 2.47 | 82.38 ± 4.23 | 90.20 ± 4.67 | 94.13 ± 6.04 | 81.19 ± 0.83 | 72.08 ± 6.14 | 72.07 ± 2.65 |
| 11. | 4k | 83.18 ± 10.33 | 77.91 ± 5.00 | 68.38 ± 11.07 | 103.04 ± 5.72 | 93.97 ± 8.24 | 70.33 ± 13.79 | 92.60 ± 2.16 | 102.06 ± 3.45 | 35.81 ± 4.22 | 120.22 ± 5.79 | 141.96 ± 10.00 | 150.46 ± 6.90 |
| 12. | 4l | n.d. | n.d. | n.d. | n.d. | n.d. | n.d. | n.d. | n.d. | n.d. | n.d. | n.d. | n.d. |
| 13. | 4m | 89.14 ± 2.30 | 93.90 ± 7.98 | 90.95 ± 2.14 | 87.89 ± 4.77 | 84.35 ± 1.62 | 81.84 ± 3.69 | 94.37 ± 1.95 | 83.17 ± 5.41 | 68.24 ± 7.67 | 167.13 ± 0.81 | 161.42 ± 10.99 | 135.98 ± 6.08 |
| 14. | 4n | 92.73 ± 3.19 | 85.55 ± 4.37 | 89.71 ± 10.43 | 101.90 ± 6.83 | 110.42 ± 3.53 | 79.94 ± 3.74 | 117.36 ± 2.96 | 84.46 ± 8.45 | 60.78 ± 4.50 | 99.76 ± 12.48 | 105.15 ± 11.46 | 68.48 ± 3.32 |
| 15. | 4o | 85.49 ± 11.10 | 83.39 ± 9.62 | 73.41 ± 4.85 | 107.06 ± 3.58 | 100.34 ± 8.79 | 72.92 ± 7.73 | 91.59 ± 4.76 | 84.62 ± 3.99 | 38.20 ± 6.03 | 111.35 ± 17.37 | 101.55 ± 11.08 | 55.56 ± 9.08 |
| 16. | 4p | 94.35 ± 7.09 | 83.34 ± 6.22 | 96.06 ± 2.34 | 92.89 ± 3.16 | 109.73 ± 4.504 | 72.54 ± 5.35 | 91.09 ± 9.17 | 94.76 ± 4.14 | 90.43 ± 8.30 | 117.56 ± 17.21 | 127.63 ± 18.70 | 91.32 ± 2.86 |
| 17. | 4q | 102.17 ± 4.77 | 95.09 ± 6.27 | 32.98 ± 1.03 | 94.55 ± 5.94 | 100.79 ± 8.68 | 50.37 ± 2.34 | 82.81 ± 4.56 | 84.34 ± 8.14 | 49.96 ± 1.45 | 122.48 ± 23.81 | 124.75 ± 4.92 | 62.09 ± 0.98 |
| 18. | 4r | 78.45 ± 3.54 | 82.10 ± 3.88 | 85.08 ± 2.17 | 94.86 ± 2.68 | 99.20 ± 3.57 | 68.99 ± 1.89 | 91.59 ± 4.76 | 84.62 ± 3.99 | 38.20 ± 6.03 | 87.77 ± 13.65 | 89.82 ± 11.28 | 54.59 ± 7.13 |
| 19. | 4s | 93.71 ± 6.65 | 100.02 ± 1.64 | 39.02 ± 9.12 | 102.68 ± 10.84 | 113.41 ± 6.07 | 72.44 ± 0.388 | 91.29 ± 6.29 | 77.23 ± 1.96 | 41.82 ± 2.96 | 82.48 ± 4.2 | 91.14 ± 11.46 | 29.87 ± 3.41 |
| 20. | 4t | 65.87 ± 4.25 | 93.87 ± 4.17 | 87.03 ± 4.83 | 106.00 ± 0.81 | 103.46 ± 5.29 | 102.87 ± 7.41 | 93.54 ± 3.84 | 98.32 ± 2.09 | 101.52 ± 3.42 | 104.43 ± 6.09 | 121.39 ± 6.08 | 70.29 ± 8.39 |
| 21. | 4u | 96.92 ± 1.46 | 85.39 ± 1.23 | 8.02 ± 1.54 | 103.74 ± 9.51 | 88.03 ± 3.17 | 18.71 ± 2.32 | 91.20 ± 2.47 | 71.81 ± 2.32 | 28.04 ± 0.99 | 104.02 ± 3.18 | 50.13 ± 6.42 | 16.64 ± 1.70 |
| 22. | 4v | 92.92 ± 4.26 | 97.98 ± 6.59 | 57.29 ± 5.64 | 99.38 ± 2.28 | 107.71 ± 10.25 | 104.73 ± 7.67 | 82.72 ± 0.91 | 89.29 ± 1.69 | 87.80 ± 3.02 | 60.38 ± 5.18 | 65.96 ± 4.84 | 75.90 ± 6.26 |
| 23. | 4w | 80.28 ± 1.46 | 74.71 ± 11.69 | 27.27 ± 2.38 | 102.04 ± 1.91 | 90.11 ± 7.77 | 38.90 ± 2.70 | 94.36 ± 0.31 | 89.72 ± 2.26 | 48.91 ± 1.17 | 103.13 ± 2.62 | 97.43 ± 9.23 | 45.60 ± 3.65 |
| 24. | 5-FU | 81.77 ± 4.89 | 57.94 ± 7.11 | 31.28 ± 5.83 | 57.38 ± 3.18 | 44.64 ± 4.80 | 18.80 ± 0.08 | 88.18 ± 1.76 | 25.62 ± 3.29 | 9.03 ± 1.29 | 67.80 ± 1.41 | 42.04 ± 2.36 | 35.71 ± 2.09 |

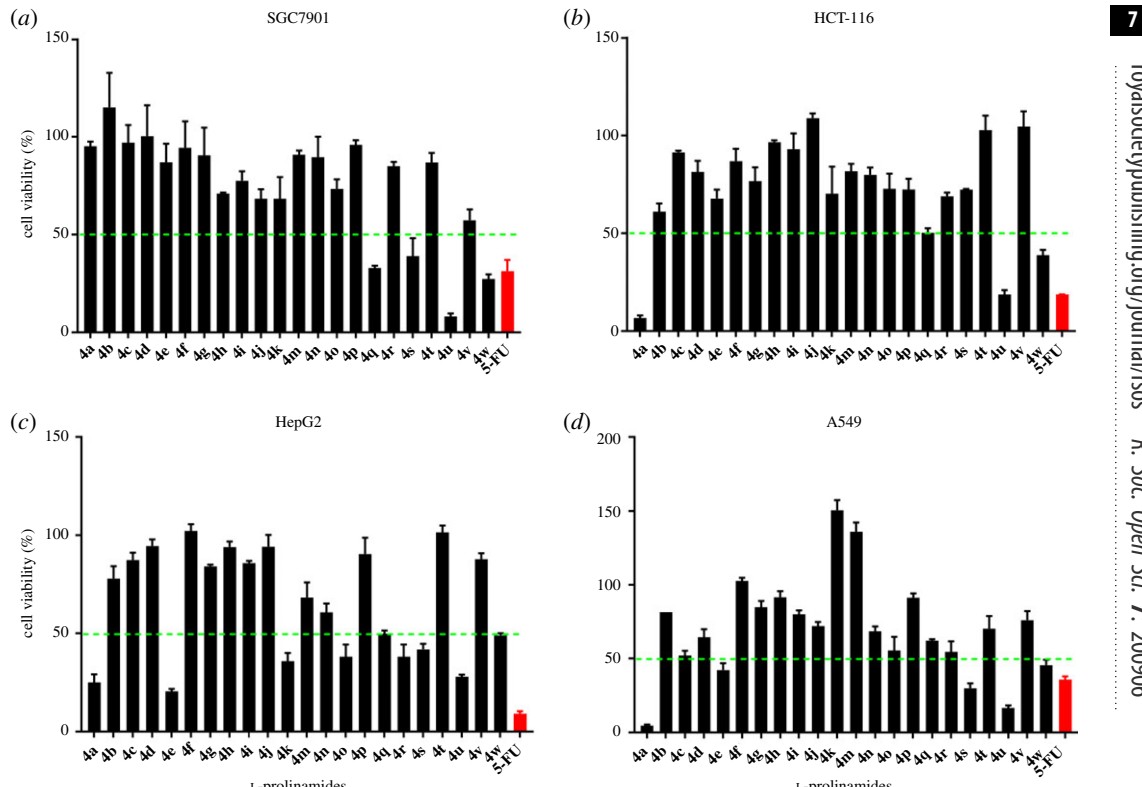

**Figure 2.** (a–d) In vitro cytotoxic activity (MTT assay) of ʟ-prolinamides (100 µM) against four human carcinoma cell lines. (a) SGC7901, human gastric carcinoma cell line; (b) HCT-116, human colon carcinoma cell line; (c) HepG2, human liver carcinoma cell line; and (d) A549, human lung carcinoma cell line. 5-FU, 5-fluorouracil (positive control). Concentration of ʟ-prolinamides, 100 µM. Data are given as mean ± s.d. (n = 3).

such as A549 (with **4f**, **4k** and **4m**), SGC7901 (with **4b** and **4d**) and HCT-116 (with **4j** and **4t**), where the percentage cell viability was greater than 100%, over the three concentrations.

The inhibitory effects of 100 µM concentrations of ʟ-prolinamides **4a**–**4w** on the four cancer cell lines under study are compared in figure 2, with the aid of bar chart diagrams. In precis, many of the prolinamides tested herein induced more than or equal to 50% cell inhibition in one or more of the cancer cell lines assayed, thereby highlighting the use of this class of compounds in cancer therapy. This is exemplified by **4a**, **4e**, **4k**, **4o**, **4q**, **4r**, **4s**, **4u** and **4w** on HepG2; **4a**, **4e**, **4s**, **4u** and **4w** on A549; **4q**, **4s**, **4u** and **4w** on SGC7901 and **4a**, **4q**, **4u** and **4w** on HCT-116. Accordingly, the human liver (HepG2) and colon (HCT-116) carcinoma cell lines proved to be the most sensitive and resistant cell lines, respectively, to the prolinamides under investigation. Additionally, ʟ-prolinamides **4u** and **4w** showed greater than 50% inhibitions for the four cell lines assayed, whereas **4a**, **4q** and **4s** were similarly active against three of the cell lines, alluding to the broad-spectrum nature of these compounds' anti-cancer activities.

It is noteworthy that excellent tumour inhibitory activities were observed with ʟ-prolinamide **4a** giving percentage inhibitions of 95.45 ± 0.67% and 93.33 ± 1.36% against A549 and HCT-116, respectively, whereas **4u** recorded 91.98 ± 1.54% (SGC7901) and 83.36 ± 1.70% (A549). These proved more potent than the control, 5-fluorouracil, with percentage inhibitions of 64.29 ± 2.09% (A549), 81.20 ± 0.08% (HCT-116) and 68.72 ± 5.83% (SGC7901), respectively. ʟ-Prolinamide **4a** was less potent against SGC7901 and HepG2 relative to the reference compound while **4u** was less potent than and comparable to 5-fluorouracil against HepG2 and HCT-116, respectively (cf. figure 2).

It is interesting to note that, across cell lines, the tertiary prolinamides, exemplified by **4a**, **4s** and **4w**, showed better anti-tumour properties than the secondary prolinamides. However, the cyclic tertiary prolinamides (**4i** and **4n**) were the exceptions. This may be owing to the restrictive nature of their carboxamide bonds. It was also observed that the longer the N′-alkyl chain substituents of the prolinamides (**4a** versus **4o**), the stronger the antiproliferative activity; in concurrence with previous reports, which suggest that the greater lipophilicity afforded by the longer chains enable greater cell uptake [31]. Besides, N′-cyclohexylprolinamide **4e** was more toxic to the HepG2 cell line than N′-phenylprolinamide **4s** and N′,N′-dicyclohexylprolinamide **4w**, whereas **4e** inhibited the A549 cell

line better than **4w** (cf. table 1 and figure 2). A plausible explanation for the aforementioned is that $N'$-cycloalkyl-substituted compounds exhibit greater antineoplastic activities than their $N'$-aryl-substituted analogues [32] and that $N',N'$-dicycloalkyl-substitution (relative to $N'$-monosubstitution) does not necessarily lead to increased antiproliferative activity.

Furthermore, it can be inferred from the results of the MTT assays of the four cell lines, except SGC7901, that branching (**4j** versus **4o**; table 1) can confer reduced inhibitory potency against cancer cell lines. This is corroborated by $N'$-doubly substituted compounds (**4j** versus **4v**) where increased branching resulted in decreased inhibitory activities against the four cell lines, except A549 (figure 2). Considering the $N'$-phenyl-containing prolinamides (e.g. **4s**; cf. tables 1 and 2), electron-withdrawing substituents on the phenyl ring (**4u**) led to increased cytotoxicity, whereas electron-donating ring substituents (**4k**) decreased cytotoxic activity. Conversely, $N'$-toluenesulfonylprolinamide **4h** showed higher cytotoxicity than $N'$-benzenesulfonylprolinamide **4f** in all assays except HCT-116. In addition, $N'$-benzylprolinamide **4q** proved more potent than $N'$-phenylprolinamide **4s** against SGC7901 and HCT-116 but less potent against HepG2 and A549 (figure 2). Moreover, the substitution of 4-H on the prolyl ring by 4-OH (**4k** versus **4m**) appeared to have had minimal effects on the cell lines, as revealed by the percentage cell viabilities of SGC7901, HCT-116 and HepG2, which were higher for **4m**; except against A549 where the converse was the case. L-Prolinamides **4k** and **4m** are also both suspected of proliferating, rather than inhibiting, the human lung carcinoma cell line (A549).

Finally, the L-prolinamides **4a–4w** synthesized and assayed in this protocol showed antiproliferative activities against one or more of the human carcinoma cell lines; with the most activity against HepG2 and the least activity against HCT-116. It is surprising, however, that none of the compounds bearing the aryl sulfonamide moiety (**4f** and **4h**) showed a more than or equal to 50% inhibition against any of the four cancer cell lines assayed since sulfonamides have been reported to exhibit substantial *in vitro* and *in vivo* anti-tumour activities [33]. It is also instructive to note that none of the prolinamides **4a–4w** exhibited a stronger cytotoxicity than 5-fluorouracil against the human liver carcinoma cell line (HepG2). The percentage cell viability of the nitrile, $N'$-(2″-cyanophenyl)-$N$-(4′-nitrophenyl)-L-prolinamide **4r** against the four carcinoma cell lines tested ranged from $38.20 \pm 6.03\%$ (against HepG2) to $99.20 \pm 3.57\%$ (against HCT-116), as shown in table 2.

# 3. Conclusion

$N'$-substituted-$N$-(4′-nitrophenyl)-L-prolinamides **4a–4w** have been successfully synthesized, characterized and assayed as potential anti-cancer agents. The synthetic route to the L-prolinamides was a facile and economical two-step protocol, starting from readily available reagents, to give the target carboxamides in 20–80% yield. The cytotoxicity of the resulting L-prolinamides were also investigated *in vitro* against the human gastric carcinoma (SGC7901), human colon carcinoma (HCT-116), human liver carcinoma (HepG2) and human lung carcinoma (A549) cell lines, using the MTT colorimetric assay.

Almost 50% of L-prolinamides **4a–4w** showed strong anti-tumour activity against HepG2 but none was as potent as the positive control, 5-fluorouracil. However, **4a** possessed stronger cancer inhibitory properties against the HCT-116 and A549 than 5-fluorouracil, whereas **4u** was comparable in antineoplastic activities to the reference compound against HCT-116 but more potent against SGC7901 and A549. Of the 23 L-prolinamides assayed, $N',N'$-dibutyl-$N$-(4′-nitrophenyl)-L-prolinamide **4a** and $N,N'$-*bis*(4′-nitrophenyl)-L-prolinamide **4u** showed the lowest percentage cell viability data across the four assayed cell lines and therefore possess the most promising cancer cells' inhibitory activities. Consequently, L-prolinamides **4a** and **4u** can be considered as potential broad-spectrum anti-cancer agents. Likewise, some of the other L-prolinamides are good lead compounds for structural optimization to develop potent antiproliferative agents for liver carcinomas.

# 4. Experimental procedure

Commercially available analytical grade reagents were used as purchased without further purification. Glassware was flame-dried and reactions carried out under an inert (dry nitrogen gas) atmosphere. Reactions were monitored by thin layer chromatography (TLC) on Merck silica gel 60 $F_{254}$ precoated plates using an ethyl acetate/$n$-hexane (1 : 2) solvent system and visualized under ultraviolet lamp (254 nm). Column chromatography was performed with silica gel (300–400 mesh) and solvents as indicated. Melting points were determined on a MEL-TEMP® capillary melting point apparatus and are

uncorrected. IR spectra were recorded on a Perkin Elmer Universal (ATR Spectrum 100) Fourier-transform-infrared spectrometer. $^1$H-NMR (400 MHz & 600 MHz) and $^{13}$C-NMR (150 MHz) spectra were recorded on a Varian-Inova (400 MHz or 600 MHz) spectrometer with CDCl$_3$ or DMSO-$d_6$ as solvent and tetramethylsilane (TMS) as internal standard. Chemical shifts ($\delta$) and coupling constants ($J$) are reported in parts per million (ppm), downfield from TMS and hertz (Hz), respectively. High-resolution mass spectra ($m/z$) were recorded on a micro TOF–QIII (ESI) spectrometer.

# 5. Chemistry

## 5.1. General procedure for the synthesis of N-(4′-substituted phenyl)-L-proline adducts

Into a clean round-bottomed flask was added L-proline **2** (12 mmol) in 1 : 1 ethanol/water (20 ml) and K$_2$CO$_3$ (3.5 equivalents), with stirring. The mixture was refluxed for 10 min before the addition of 4-substituted fluorobenzene **1** (10 mmol). The reaction mixture was then refluxed for 3h before cooling to ambient temperature, concentrating *in vacuo* and extracting into CH$_2$Cl$_2$ (10 ml). The aqueous layer was then acidified with 6$M$ HCl and extracted into CH$_2$Cl$_2$ (20 ml × 3). The resulting organic layer was then washed with saturated brine solution (20 ml), dried over anhydrous Na$_2$SO$_4$, filtered and concentrated, under pressure, to give **3** as a solid.

## 5.2. Spectroscopic data

***N*-(4′-nitrophenyl)-L-proline (3a)**: yellow crystals (2.56 g, 90%); m.p. 160–161°C; IR (neat) $v_{max}$ (cm$^{-1}$): 3050, 2965, 2871, 1714, 1597, 1513, 1304, 1186, 1111; $^1$H-NMR (400 MHz, CDCl$_3$) $\delta$: 10.52 (1H, s, –COO**H**), 8.10 (2H, d, $J$ = 9.1 Hz, Ar**H**), 6.49 (2H, d, $J$ = 9.1 Hz, Ar**H**), 4.39 (1H, d, $J$ = 7.0 Hz, –C**H**COOH), 3.69–3.55 (1H, m, –C**H**$_a$H$_b$N–), 3.47 (1H, dd, $J$ = 16.8, 8.4 Hz, –CH$_a$**H**$_b$N–), 2.47–2.28 (2H, m, –CH$_2$C**H**$_2$–), 2.25–2.09 (2H, m, –C**H**$_2$CH$_2$–); $^{13}$C-NMR (150 MHz, CDCl$_3$) $\delta$: 178.4 (–**C**OOH), 151.0; 137.6; 126.2; 111.1 (Ar**H**), 60.5 (–**C**HCOOH), 48.6 (–**C**H$_2$N–), 30.8 (–**C**H$_2$CH$_2$–), 23.6 (–**C**H$_2$CH$_2$–); HRMS (ESI) calculated for C$_{11}$H$_{12}$N$_2$O$_4$ [M + H]$^+$ 237.0875, found 237.0879.

***N*-(4-cyanophenyl)-L-proline (3b)**: off-white crystals (1.79 g, 69%); m.p. 151–152°C; IR (neat) $v_{max}$ (cm$^{-1}$): 3463, 3040, 2949, 2857, 2208, 1727, 1600, 1215, 1172; $^1$H-NMR (400 MHz, CDCl$_3$) $\delta$: 10.31 (1H, s, –COO**H**), 7.45 (2H, d, $J$ = 8.1 Hz, Ar**H**), 6.52 (2H, d, $J$ = 8.1 Hz, Ar**H**), 4.30 (1H, d, $J$ = 7.9 Hz, –C**H**COOH), 3.56 (1H, d, $J$ = 5.9 Hz, –C**H**$_a$H$_b$N–), 3.40 (1H, dd, $J$ = 16.2, 8.0 Hz, –CH$_a$**H**$_b$N–), 2.40–2.30 (2H, m, –CH$_2$C**H**$_2$–), 2.22–2.12 (2H, m, –C**H**$_2$CH$_2$–); $^{13}$C-NMR (150 MHz, CDCl$_3$) $\delta$: 178.6 (–**C**OOH), 149.3; 133.8 (Ar**H**), 120.5 (–**C**N), 112.2; 98.7 (Ar**H**), 60.4 (–**C**HCOOH), 48.4 (–**C**H$_2$N–), 30.9 (–**C**H$_2$CH$_2$–), 23.7 (–**C**H$_2$CH$_2$–); HRMS (ESI) calculated for C$_{12}$H$_{12}$N$_2$O$_2$ [M + H]$^+$ 217.0977, found 217.0991.

**4-Hydroxy-*N*-(4′-nitrophenyl)-L-proline (3c)**: yellow crystals (2.69 g, 89%); m.p. 152–153°C; IR (neat) $v_{max}$ (cm$^{-1}$): 3352, 3095, 2939, 1734, 1580, 1511, 1283, 1172, 1104, 1075; $^1$H-NMR (400 MHz, DMSO-$d_6$) $\delta$: 8.59 (2H, d, $J$ = 9.0 Hz, Ar**H**), 7.09 (2H, d, $J$ = 9.0 Hz, Ar**H**), 5.82 (1H, s, *prolyl* 4-O**H**), 4.99 (2H, s, –C**H**COOH; –C**H**OH), 4.23–4.19 (1H, m, –C**H**$_a$H$_b$N–), 3.89–3.86 (1H, m, –CH$_a$**H**$_b$N–), 2.84–2.81 (1H, m, –CHC**H**$_a$H$_b$–), 2.77–2.50 (1H, m, –CHCH$_a$**H**$_b$–); $^{13}$C-NMR (150 MHz, DMSO-$d_6$) $\delta$: 173.7 (–**C**OOH), 151.9; 136.5; 126.0; 111.4 (Ar**H**), 68.2 (–**C**HCOOH), 59.5 (–**C**HOH), 56.8 (–**C**H$_2$N–), 38.9 (–CHC**H**$_2$–); HRMS (ESI) calculated for C$_{11}$H$_{12}$N$_2$O$_5$ [M + H]$^+$ 253.0824, found 253.0825.

## 5.3. General procedure for the synthesis of N-(4′-substituted phenyl)-L-prolinamides

An oven-dried, N$_2$-evacuated round-bottomed flask equipped with a magnetic stirring bar was charged with *N*-(4′-substituted phenyl)-L-proline **3** (5 mmol) in CH$_2$Cl$_2$ (20 ml), SOCl$_2$ (2 ml, 27.6 mmol) and dimethyl formamide (one drop) and stirred for 3–4 h at ambient temperature. The reaction mixture was then concentrated to the crude acid chloride **5**, which was taken up in CH$_2$Cl$_2$ (20 ml) and concentrated *in vacuo* to remove excess SOCl$_2$. The acid chloride was dissolved in CH$_2$Cl$_2$ (20 ml) and cooled to 0°C, under N$_2$, and NEt$_3$ (1.5 equivalents) was added. The amine (1.5 equivalents) was then added, with the aid of a dropping funnel and the reaction was left to stir overnight at ambient temperature. To work-up, the reaction mixture was acidified with 2$M$ HCl (pH = 2) and the organic layer was washed with saturated brine solution (10 ml), dried over anhydrous Na$_2$SO$_4$, filtered and concentrated under vacuum to a paste, which was purified via column chromatography on silica gel using *n*-hexane and ethyl acetate (2 : 1) as mobile phase.

## 5.4. Spectroscopic data

**N′,N′-dibutyl-N-(4′-nitrophenyl)-ʟ-prolinamide (4a)**: dark orange solid (1.04 g, 60%); m.p. 74–75°C; IR (neat) $v_{max}$ (cm$^{-1}$): 3090, 2954, 2928, 2870, 2852, 1638, 1596, 1579, 1518, 1484, 1296, 1108; $^1$H-NMR (400 MHz, CDCl$_3$), Rotamers 1:1, δ: 8.08 (2H, d, J = 9.1 Hz, Ar<u>H</u>), 6.36 (2H, d, J = 8.6 Hz, Ar<u>H</u>), 4.56 (1H, d, J = 7.0 Hz, –NC<u>H</u>CON′–), 3.69 (1H, td, J = 9.2, 3.3 Hz, –C<u>H</u>$_a$H$_b$N–), 3.52 (1H, d, J = 8.7 Hz, –CH$_a$<u>H</u>$_b$N–), 3.46–3.36 (2H, m, –N′C<u>H</u>$_2$(CH$_2$)$_2$CH$_3$), 3.30 (2H, dd, J = 13.5, 6.9 Hz, –N′C<u>H</u>$_2$(CH$_2$)$_2$CH$_3$), 2.41–2.33 (m, 1H, –CH$_2$C<u>H</u>$_a$H$_b$CH–), 2.22 (1H, d, J = 7.6 Hz, –CH$_2$CH$_a$<u>H</u>$_b$CH–), 2.14–2.04 (2H, m, –C<u>H</u>$_2$CH$_a$H$_b$CH–), 1.71 (2H, dt, J = 15.5, 7.7 Hz, –N′CH$_2$C<u>H</u>$_2$CH$_2$CH$_3$), 1.55–1.37 (4H, m, –N′CH$_2$C<u>H</u>$_2$CH$_2$CH$_3$; –N′(CH$_2$)$_2$C<u>H</u>$_2$CH$_3$), 1.28 (2H, dd, J = 15.7, 8.2 Hz, –N′(CH$_2$)$_2$C<u>H</u>$_2$CH$_3$), 1.02 (3H, t, J = 7.3 Hz, –N′(CH$_2$)$_3$C<u>H</u>$_3$), 0.91 (3H, t, J = 7.3 Hz, –N′(CH$_2$)$_3$C<u>H</u>$_3$); $^{13}$C-NMR (150 MHz, CDCl$_3$), Rotamers 1:1, δ: 170.6 (–<u>C</u>ON′–), 151.3; 137.6; 126.2; 110.8 (<u>Ar</u>H), 59.1 (–<u>C</u>HCON′–), 49.0 (–<u>C</u>H$_2$N–), 47.3; 46.0 (–N′<u>C</u>H$_2$(CH$_2$)$_2$CH$_3$), 31.4; 31.1 (–N′CH$_2$<u>C</u>H$_2$CH$_2$CH$_3$), 29.6 (–<u>C</u>H$_2$CH$_2$CH–), 23.4 (–<u>C</u>H$_2$CH$_2$CH–), 20.21; 20.15 (–N′(CH$_2$)$_2$<u>C</u>H$_2$CH$_3$), 13.82; 13.78 (–N′(CH$_2$)$_3$<u>C</u>H$_3$); HRMS (ESI) calculated for C$_{19}$H$_{29}$N$_3$O$_3$ [M + H]$^+$ 348.2287, found 348.2282.

**N′-butyl-N-(4′-nitrophenyl)-ʟ-prolinamide (4b)**: yellow solid (0.70 g, 48%); m.p. 145–147°C; IR (neat) $v_{max}$ (cm$^{-1}$): 3280, 3085, 2930, 2864, 1650, 1597, 1514, 1484, 1305, 1240, 1188, 1111; $^1$H-NMR (400 MHz, CDCl$_3$) δ: 8.07 (2H, d, J = 9.3 Hz, Ar<u>H</u>), 6.54 (2H, d, J = 9.3 Hz, Ar<u>H</u>), 6.13 (1H, t, J = 5.3 Hz, –CON<u>H</u>), 4.18–4.14 (1H, m, –NC<u>H</u>CON′–), 3.71 (1H, ddd, J = 9.7, 7.7, 2.1 Hz, –N′C<u>H</u>$_a$H$_b$(CH$_2$)$_2$CH$_3$), 3.38 (1H, td, J = 9.9, 6.7 Hz, –N′CH$_a$<u>H</u>$_b$(CH$_2$)$_2$CH$_3$), 3.23 (2H, td, J = 7.2, 1.7 Hz, –C<u>H</u>$_2$N–), 2.35–2.27 (2H, m, –CH$_2$C<u>H</u>$_2$CH–), 2.16–1.98 (2H, m, –C<u>H</u>$_2$CH$_2$CH–), 1.45–1.36 (2H, m, –N′CH$_2$C<u>H</u>$_2$CH$_2$CH$_3$), 1.24 (2H, t, J = 7.3 Hz, –N′(CH$_2$)$_2$C<u>H</u>$_2$CH$_3$), 0.85 (3H, t, J = 7.3 Hz, –N′(CH$_2$)$_3$C<u>H</u>$_3$); $^{13}$C-NMR (150 MHz, CDCl$_3$) δ: 171.7 (–<u>C</u>ON′–), 151.7; 138.3; 126.0; 111.7 (<u>Ar</u>H), 64.0 (–<u>C</u>HCON′–), 49.5 (–<u>C</u>H$_2$N–), 39.2 (–N′<u>C</u>H$_2$(CH$_2$)$_2$CH$_3$), 31.5 (–N′CH$_2$<u>C</u>H$_2$CH$_2$CH$_3$), 31.4 (–CH$_2$<u>C</u>H$_2$CH–), 23.9 (–<u>C</u>H$_2$CH$_2$CH–), 19.9 (–N′(CH$_2$)$_2$<u>C</u>H$_2$CH$_3$), 13.6 (–N′(CH$_2$)$_3$<u>C</u>H$_3$); HRMS (ESI) calculated for C$_{15}$H$_{21}$N$_3$O$_3$ [M + Na]$^+$ 314.1481, found 314.1485.

**N-(4′-nitrophenyl)-N′-propyl-ʟ-prolinamide (4c)**: orange solid (0.68 g, 49%); m.p. 179–180°C; IR (neat) $v_{max}$ (cm$^{-1}$): 3277, 3084, 2963, 2928, 2871, 1649, 1597, 1554, 1513, 1483, 1305, 1238, 1186, 1111; $^1$H-NMR (400 MHz, CDCl$_3$) δ: 8.06 (2H, d, J = 9.3 Hz, Ar<u>H</u>), 6.54 (2H, d, J = 9.3 Hz, Ar<u>H</u>), 6.18 (1H, s, –CON<u>H</u>), 4.20–4.12 (1H, m, –NC<u>H</u>CON′–), 3.77–3.66 (1H, m, –N′C<u>H</u>$_a$H$_b$CH$_2$CH$_3$), 3.38 (1H, td, J = 9.8, 6.8 Hz, –N′CH$_a$<u>H</u>$_b$CH$_2$CH$_3$), 3.27–3.13 (2H, m, –C<u>H</u>$_2$N–), 2.35–2.26 (2H, m, –CH$_2$C<u>H</u>$_2$CH–), 2.16–1.96 (2H, m, –C<u>H</u>$_2$CH$_2$CH–), 1.50–1.38 (2H, m, N′CH$_2$C<u>H</u>$_2$CH$_3$), 0.81 (3H, t, J = 7.4 Hz, N′CH$_2$CH$_2$C<u>H</u>$_3$); $^{13}$C-NMR (150 MHz, CDCl$_3$) δ: 171.7 (–<u>C</u>ON′–), 151.6; 138.4; 126.0; 111.8 (<u>Ar</u>H), 64.0 (–<u>C</u>HCON′–), 49.5 (–<u>C</u>H$_2$N–), 41.1 (–N′<u>C</u>H$_2$CH$_2$CH$_3$), 31.5 (–CH$_2$<u>C</u>H$_2$CH–), 23.9 (–<u>C</u>H$_2$CH$_2$CH–), 22.8 (–N′CH$_2$<u>C</u>H$_2$CH$_3$), 11.2 (–N′CH$_2$CH$_2$<u>C</u>H$_3$); HRMS (ESI) calculated for C$_{14}$H$_{19}$N$_3$O$_3$ [M + H]$^+$ 278.1505, found 278.1503.

**N-(4′-nitrophenyl)-N′-(1″-prop-1″-ynyl)-ʟ-prolinamide (4d)**: yellow solid (0.74 g, 54%); m.p. 158–159°C; IR (neat) $v_{max}$ (cm$^{-1}$): 3289, 3070, 2974, 2924, 2870, 1651, 1578, 1535, 1511, 1477, 1432, 1290, 1231, 1156, 1113; $^1$H-NMR (400 MHz, CDCl$_3$) δ: 8.04 (2H, d, J = 9.3 Hz, ArH), 6.52 (2H, d, J = 9.3 Hz, ArH), 6.47 (1H, t, J = 5.3 Hz, –CON<u>H</u>), 4.21–4.16 (1H, m, –NC<u>H</u>CON′–), 4.06–4.01 (2H, m, –C<u>H</u>$_2$N–), 3.74 (1H, ddd, J = 9.6, 7.6, 2.2 Hz, –CH$_2$C<u>H</u>$_a$H$_b$CH–), 3.38 (1H, td, J = 9.8, 6.8 Hz, –CH$_2$CH$_a$<u>H</u>$_b$CH–), 2.36–2.28 (2H, m, propynyl –C<u>H</u>$_2$-H), 2.18 (1H, t, J = 2.5 Hz, propynyl –CH$_2$-<u>H</u>), 2.15–2.00 (2H, m, –C<u>H</u>$_2$CH$_a$H$_b$CH–); $^{13}$C-NMR (150 MHz, CDCl$_3$) δ: 171.8 (–<u>C</u>ON′–), 151.6; 138.6; 126.1; 111.8 (<u>Ar</u>H), 79.0 (N′<u>C</u>CCH$_3$), 71.6 (–<u>C</u>HCON′–), 63.8 (N′C<u>C</u>CH$_3$), 49.5 (–<u>C</u>H$_2$N–), 31.4 (–CH$_2$<u>C</u>H$_2$CH–), 29.1 (–<u>C</u>H$_2$CH$_2$CH–), 23.9 (N′CC<u>C</u>H$_3$); HRMS (ESI) calculated for C$_{14}$H$_{15}$N$_3$O$_3$ [M + Na]$^+$ 296.1011, found 296.0984.

**N′-cyclohexyl-N-(4′-nitrophenyl)-ʟ-prolinamide (4e)**: pale yellow solid (0.68 g, 43%); m.p. 252–253°C; IR (neat) $v_{max}$ (cm$^{-1}$): 3267, 3084, 2925, 2853, 1649, 1598, 1515, 1485, 1440, 1309, 1190, 1113; $^1$H-NMR (400 MHz, CDCl$_3$) δ: 8.06 (2H, d, J = 9.3 Hz, Ar<u>H</u>), 6.53 (2H, d, J = 9.3 Hz, Ar<u>H</u>), 5.94 (1H, d, J = 8.4 Hz, –CON<u>H</u>), 4.12 (1H, dd, J = 7.8, 3.7 Hz, –NC<u>H</u>CON′–), 3.73 (2H, dddd, J = 11.9, 9.7, 7.3, 3.3 Hz, –HN′C<u>H</u>–), 3.37 (1H, td, J = 9.8, 6.7 Hz, –C<u>H</u>$_2$N–), 2.36–2.24 (2H, m, prolyl –CH$_2$C<u>H</u>$_2$CH–), 2.15–1.97 (2H, m, cyclohexyl –C<u>H</u>$_2$–), 1.88–1.72 (2H, m, prolyl –C<u>H</u>$_2$CH$_2$CH–), 1.59 (3H, m, cyclohexyl –C<u>H</u>$_2$–), 1.37–1.26 (2H, m, cyclohexyl –C<u>H</u>$_2$–), 1.12–0.92 (3H, m, cyclohexyl –C<u>H</u>$_2$–); $^{13}$C-NMR (150 MHz, CDCl$_3$), Rotamers (1:1), δ: 170.7 (–<u>C</u>ON′–), 151.6; 138.4; 126.0; 111.8 (<u>Ar</u>H), 64.0 (–<u>C</u>HCON′–), 49.5 (–<u>C</u>H$_2$N–), 48.2 (–HN′<u>C</u>H–), 33.0; 32.8 (cyclohexyl –CH<u>C</u>H$_2$–), 31.5 (prolyl –CH$_2$<u>C</u>H$_2$CH–), 29.6 (cyclohexyl –CH$_2$<u>C</u>H$_2$CH$_2$–), 25.3 (prolyl –<u>C</u>H$_2$CH$_2$CH–), 24.8; 24.7 (cyclohexyl –<u>C</u>H$_2$CH$_2$CH$_2$–); HRMS (ESI) calculated for C$_{17}$H$_{23}$N$_3$O$_3$ [M + H]$^+$ 318.1818, found 318.1794.

**N-(4′-nitrophenyl)-N′-(phenylsulfonyl)-ʟ-prolinamide (4f)**: yellow solid (0.80 g, 58%); m.p. 189–191°C; IR (neat) $v_{max}$ (cm$^{-1}$): 3226, 3080, 2990, 2880, 1716, 1601, 1509, 1479, 1410, 1313, 1115; $^1$H-NMR (400 MHz, CDCl$_3$) δ: 9.13 (1H, s, –CON<u>H</u>–), 7.85 (2H, d, J = 9.3 Hz, Ar<u>H</u>), 7.81–7.76 (2H, m, Ar<u>H</u>), 7.70 (1H, t, J = 7.5 Hz, Ar<u>H</u>), 7.50 (2H, t, J = 7.9 Hz, Ar<u>H</u>), 6.24 (2H, d, J = 9.2 Hz, Ar<u>H</u>), 4.03 (1H, dd, J = 9.2,

3.2 Hz, –NCHCON′–), 3.81–3.75 (1H, m, –C$\underline{H}_a$H$_b$N–), 3.31 (1H, dd, $J$ = 16.5, 8.9 Hz, –CH$_a$$\underline{H}_b$N–), 2.39–2.30 (1H, m, –CH$_2$C$\underline{H}_a$H$_b$CH–), 2.26–2.18 (1H, m, –CH$_2$CH$_a$$\underline{H}_b$CH–), 2.12–2.05 (2H, m, –C$\underline{H}_2$CH$_a$H$_b$CH–); $^{13}$C-NMR (150 MHz, CDCl$_3$) $\delta$: 171.4 (–$\underline{C}$ON′–), 150.9; 138.9; 137.5; 134.5; 129.0; 128.0; 125.8; 111.8 (Ar$\underline{C}$H), 63.9 (–$\underline{C}$HCON′–), 49.6 (–$\underline{C}$H$_2$N–), 31.3 (–$\underline{C}$H$_2$CH$_2$CH–), 24.0 (–CH$_2$$\underline{C}$H$_2$CH–); HRMS (ESI) calculated for C$_{17}$H$_{17}$N$_3$O$_5$S [M + Na]$^+$ 398.0787, found 398.0775.

**N′-(tert-butyl)-N-(4′-nitrophenyl)-ʟ-prolinamide (4g)**: yellow solid (1.05 g, 72%); m.p. 141–142°C; IR (neat) $\nu_{max}$ (cm$^{-1}$): 3296, 3052, 2967, 1657, 1595, 1548, 1518, 1480, 1396, 1296, 1193, 1108; $^1$H-NMR (400 MHz, CDCl$_3$) $\delta$: 8.07 (2H, d, $J$ = 9.3 Hz, Ar$\underline{H}$), 6.55 (2H, d, $J$ = 9.3 Hz, Ar$\underline{H}$), 5.97 (1H, s, –CON$\underline{H}$), 4.06 (1H, dd, $J$ = 8.7, 2.8 Hz, –NC$\underline{H}$CON′–), 3.75–3.69 (1H, m, –C$\underline{H}_a$H$_b$N–), 3.40 (1H, dt, $J$ = 16.5, 8.4 Hz, –CH$_a$$\underline{H}_b$N–), 2.36–2.22 (2H, m, –CH$_2$C$\underline{H}_2$CH–), 2.11 (2H, td, $J$ = 6.8, 3.5 Hz, –C$\underline{H}_2$CH$_2$CH–), 1.31 (9H, s, $t$-butyl –C$\underline{H}_3$); $^{13}$C-NMR (150 MHz, CDCl$_3$) $\delta$: 170.9 (–$\underline{C}$ON′–), 151.6; 138.6; 126.0; 111.8 (Ar$\underline{C}$H), 64.6 (–$\underline{C}$HCON′–), 51.3 (–N′$\underline{C}$(CH$_3$)$_3$), 49.5 (–$\underline{C}$H$_2$N–), 31.4 (–$\underline{C}$H$_2$CH$_2$CH–), 28.6 (–CH$_2$$\underline{C}$H$_2$CH–), 23.9 (–N′C($\underline{C}$H$_3$)$_3$); HRMS (ESI) calculated for C$_{15}$H$_{21}$N$_3$O$_3$ [M + Na]$^+$ 314.1481, found. 314.1463.

**N-(4′-nitrophenyl)-N′-(4″-tosyl)-ʟ-prolinamide (4h)**: yellow solid (0.99 g, 51%); m.p. 93–95°C; IR (neat) $\nu_{max}$ (cm$^{-1}$): 3237, 3025, 2905, 2872, 1719, 1596, 1492, 1296, 1172, 1110, 1083; $^1$H-NMR (600 MHz, CDCl$_3$) $\delta$: 9.34 (1H, s, –CON$\underline{H}$), 7.76 (2H, d, $J$ = 9.1 Hz, Ar$\underline{H}$), 7.57 (2H, d, $J$ = 8.2 Hz, Ar$\underline{H}$), 7.24 (2H, d, $J$ = 8.1 Hz, Ar$\underline{H}$), 6.18 (2H, d, $J$ = 9.0 Hz, Ar$\underline{H}$), 4.01 (1H, dd, $J$ = 9.2, 3.0 Hz, –NC$\underline{H}$CON′–), 3.79–3.74 (1H, m, –C$\underline{H}_a$H$_b$N–), 3.28 (1H, q, $J$ = 8.3 Hz, –CH$_a$$\underline{H}_b$N–), 2.47 (3H, s, tosyl –C$\underline{H}_3$), 2.37–2.31 (1H, m, –CH$_2$C$\underline{H}_a$H$_b$CH–), 2.20 (1H, ddd, $J$ = 13.1, 8.3, 4.9 Hz, –CH$_2$CH$_a$$\underline{H}_b$CH–), 2.10–2.04 (2H, m, –C$\underline{H}_2$CH$_a$H$_b$CH–); $^{13}$C-NMR (150 MHz, CDCl$_3$) $\delta$: 171.6 (–$\underline{C}$ON′–), 150.9; 146.0; 138.7; 134.5; 129.5; 128.0; 125.7; 111.7 (Ar$\underline{C}$H), 63.8 (–$\underline{C}$HCON′–), 49.5 (–$\underline{C}$H$_2$N–), 31.4 (–$\underline{C}$H$_2$CH$_2$CH–), 24.0 (–CH$_2$$\underline{C}$H$_2$CH–), 21.7 (Ar$\underline{C}$H$_3$); HRMS (ESI) calculated for C$_{18}$H$_{19}$N$_3$O$_5$S [M + Na]$^+$ 412.0948, found 412.0938.

**4″-Morpholinyl N-(4′-nitrophenyl)-2-pyrrolidinyl ketone (4i)**: yellow solid (0.69 g, 45%); m.p. 214–216°C; IR (neat) $\nu_{max}$ (cm$^{-1}$): 2913, 2886, 2863, 2813, 1639, 1595, 1517, 1476, 1433, 1287, 1234, 1199, 1107, 1034; $^1$H-NMR (400 MHz, CDCl$_3$) $\delta$: 8.07 (2H, d, $J$ = 9.4 Hz, Ar$\underline{H}$), 6.35 (2H, d, $J$ = 9.2 Hz, Ar$\underline{H}$), 4.61 (1H, dd, $J$ = 8.6, 2.4 Hz, –NC$\underline{H}$CON′–), 3.85–3.48 (10H, m, morpholinyl –C$\underline{H}_2$–; –C$\underline{H}_2$N–), 2.42–2.31 (1H, m, –CH$_2$C$\underline{H}_a$H$_b$CH–), 2.23–2.02 (3H, m, –CH$_2$CH$_a$$\underline{H}_b$CH–; –C$\underline{H}_2$CH$_a$H$_b$CH–); $^{13}$C-NMR (150 MHz, CDCl$_3$) $\delta$: 169.6 (–$\underline{C}$ON′–), 151.1; 137.4; 126.2; 110.9 (Ar$\underline{C}$H), 67.0; 66.5 (–$\underline{C}$H$_2$O–), 58.9 (–$\underline{C}$HCON′–), 48.8 (–$\underline{C}$H$_2$N–), 45.8; 42.5 (–N′$\underline{C}$H$_2$–), 30.5 (–$\underline{C}$H$_2$CH$_2$CH–), 23.4 (–CH$_2$$\underline{C}$H$_2$CH–); HRMS (ESI) calculated for C$_{15}$H$_{19}$N$_3$O$_4$ [M + Na]$^+$ 328.1273, found 328.1274.

**N′,N′-diisopropyl-N-(4′-nitrophenyl)-ʟ-prolinamide (4j)**: yellow solid (0.45 g, 28%); m.p. 159–160°C; IR (neat) $\nu_{max}$ (cm$^{-1}$): 3087, 3015, 1596, 1507, 1309, 1111; $^1$H-NMR (400 MHz, CDCl$_3$), Rotamers (1:1), $\delta$: 8.06 (2H, d, $J$ = 8.8 Hz, Ar$\underline{H}$), 6.34 (2H, d, $J$ = 7.0 Hz, Ar$\underline{H}$), 4.52 (1H, d, $J$ = 8.0 Hz, –NC$\underline{H}$CON′–), 4.11 (1H, dt, $J$ = 12.7, 6.2 Hz, –N′C$\underline{H}$(CH$_3$)$_2$), 3.65 (1H, d, $J$ = 5.4 Hz, N′C$\underline{H}$(CH$_3$)$_2$), 3.52 (1H, dd, $J$ = 16.2, 8.0 Hz, –C$\underline{H}_a$H$_b$N–), 3.43 (1H, dt, $J$ = 12.7, 6.2 Hz, –CH$_a$$\underline{H}_b$N–), 2.41–2.29 (1H, m, –CH$_2$C$\underline{H}_a$H$_b$CH–), 2.21 (1H, dt, $J$ = 16.4, 8.1 Hz, –CH$_2$CH$_a$$\underline{H}_b$CH–), 2.12–1.98 (2H, m, –C$\underline{H}_2$CH$_a$H$_b$CH–), 1.43–1.23 (12H, m, –N′CH(C$\underline{H}_3$)$_2$); $^{13}$C-NMR (150 MHz, CDCl$_3$), Rotamers (4:3), $\delta$: 169.3 (–$\underline{C}$ON′–), 151.4; 137.0; 126.1; 110.7 (Ar$\underline{C}$H), 60.2 (–$\underline{C}$HCON′–), 48.9; 48.4 (–N′$\underline{C}$H(CH$_3$)$_2$), 46.4 (–$\underline{C}$H$_2$N–), 30.7 (–$\underline{C}$H$_2$CH$_2$CH–), 23.4 (–CH$_2$$\underline{C}$H$_2$CH–), 21.2; 20.6; 20.53; 20.45 (–N′CH($\underline{C}$H$_3$)$_2$); HRMS (ESI) calculated for C$_{17}$H$_{25}$N$_3$O$_3$ [M + H]$^+$ 320.1974, found 320.1969.

**N-(4′-nitrophenyl)-N′-(4″-tolyl)-ʟ-prolinamide (4k)**: yellow solid (1.39 g, 80%); m.p. 218–220°C; IR (neat) $\nu_{max}$ (cm$^{-1}$): 3291, 2913, 2958, 1667, 1598, 1537, 1513, 1480, 1297, 1182, 1111; $^1$H-NMR (400 MHz, CDCl$_3$) $\delta$: 8.06 (1H, s, –CON$\underline{H}$), 8.01 (2H, d, $J$ = 9.2 Hz, Ar$\underline{H}$), 7.33 (2H, d, $J$ = 8.4 Hz, Ar$\underline{H}$), 7.07 (2H, d, $J$ = 8.2 Hz, Ar$\underline{H}$), 6.57 (2H, d, $J$ = 9.3 Hz, Ar$\underline{H}$), 4.24 (1H, t, $J$ = 5.8 Hz, –NC$\underline{H}$CON′–), 3.83–3.76 (1H, m, –C$\underline{H}_a$H$_b$N–), 3.40 (1H, dd, $J$ = 17.3, 8.8 Hz, –CH$_a$$\underline{H}_b$N–), 2.41–2.33 (2H, m, –CH$_2$C$\underline{H}_2$CH–), 2.28 (3H, s, Ar$\underline{C}$H$_3$), 2.14 (2H, dd, $J$ = 13.8, 7.0 Hz, –C$\underline{H}_2$CH$_2$CH–); $^{13}$C-NMR (150 MHz, CDCl$_3$) $\delta$: 170.3 (–$\underline{C}$ON′–), 151.7; 138.8; 134.8; 134.2; 129.5; 126.1; 120.4; 112.0 (Ar$\underline{C}$H), 64.5 (–$\underline{C}$HCON′–), 49.8 (–$\underline{C}$H$_2$N–), 31.6 (–$\underline{C}$H$_2$CH$_2$CH–), 24.0 (–CH$_2$$\underline{C}$H$_2$CH–), 20.9 (Ar$\underline{C}$H$_3$); HRMS (ESI) calculated for C$_{18}$H$_{19}$N$_3$O$_3$ [M + Na]$^+$ 348.1324, found 348.1341.

**N-(4′-cyanophenyl)-N′-(4″-tolyl)-ʟ-prolinamide (4l)**: brown oil (0.34 g, 21%); IR (neat) $\nu_{max}$ (cm$^{-1}$): 3303, 2920, 2868, 2211, 1666, 1602, 1511, 1364, 1309, 1174; $^1$H-NMR (600 MHz, CDCl$_3$) $\delta$: 7.93 (1H, s, –CON$\underline{H}$), 7.48 (2H, d, $J$ = 8.8 Hz, Ar$\underline{H}$), 7.32 (2H, d, $J$ = 8.4 Hz, Ar$\underline{H}$), 7.09 (2H, d, $J$ = 8.2 Hz, Ar$\underline{H}$), 6.67 (2H, d, $J$ = 8.8 Hz, Ar$\underline{H}$), 4.17 (1H, dd, $J$ = 8.6, 3.1 Hz, –NC$\underline{H}$CON′–), 3.80–3.76 (1H, m, –C$\underline{H}_a$H$_b$N–), 3.36 (1H, td, $J$ = 9.7, 6.8 Hz, –CH$_a$$\underline{H}_b$N–), 2.41–2.33 (2H, m, –CH$_2$C$\underline{H}_2$CH–), 2.29 (3H, s, Ar$\underline{C}$H$_3$), 2.16–2.05 (2H, m, –C$\underline{H}_2$CH$_2$CH–); $^{13}$C-NMR (150 MHz, CDCl$_3$) $\delta$: 170.6 (–$\underline{C}$ON′–), 150.0; 134.6; 134.3; 133.7; 129.5 (Ar$\underline{C}$H), 120.3 (–$\underline{C}$N), 113.2; 100.4 (Ar$\underline{C}$H), 64.6 (–$\underline{C}$HCON′–), 49.6 (–$\underline{C}$H$_2$N–), 31.6 (–$\underline{C}$H$_2$CH$_2$CH–), 24.0 (–CH$_2$$\underline{C}$H$_2$CH–), 20.9 (Ar$\underline{C}$H$_3$); HRMS (ESI) calculated for C$_{19}$H$_{19}$N$_3$O [M + Na]$^+$ 328.1426, found 328.1423.

***trans*-4-hydroxy-*N*-(4′-nitrophenyl)-*N*′-(4″-tolyl)-ʟ-prolinamide (4m)**: yellow solid (0.36 g, 21%); m.p. 236–238°C; IR (neat) $v_{max}$ (cm$^{-1}$): 3297, 2952, 2918, 1659, 1596, 1493, 1311, 1196, 1113; $^{1}$H-NMR (600 MHz, DMSO-$d_6$) δ: 9.26 (1H, s, –CON<u>H</u>), 7.11 (2H, d, *J* = 9.4 Hz, Ar<u>H</u>), 6.50 (2H, d, *J* = 8.4 Hz, Ar<u>H</u>), 6.15 (2H, d, *J* = 8.3 Hz, Ar<u>H</u>), 5.65 (2H, d, *J* = 8.6 Hz, Ar<u>H</u>), 4.40 (1H, s, –C<u>H</u>OH), 3.62 (1H, t, *J* = 7.4 Hz, –NC<u>H</u>CON′–), 3.56 (1H, dd, *J* = 7.5, 3.9 Hz, –C<u>H</u>OH), 2.84 (1H, dd, *J* = 10.7, 4.8 Hz, –C<u>H</u>$_a$H$_b$N–), 1.56–1.55 (1H, m, –CH$_a$<u>H</u>$_b$N–), 1.44–1.39 (1H, m, –CHC<u>H</u>$_a$H$_b$CH–), 1.29 (3H, s, ArC<u>H</u>$_3$), 1.25–1.20 (1H, m, –CHCH$_a$<u>H</u>$_b$CH–); $^{13}$C-NMR (150 MHz, DMSO-$d_6$) δ: 170.5 (–<u>C</u>ON′–), 152.3; 136.6; 136.4; 133.2; 129.5; 126.3; 120.1; 111.7 (<u>Ar</u>H), 68.6 (–<u>C</u>HCON′–), 61.6 (–<u>C</u>HOH), 57.8 (–<u>C</u>H$_2$N–), 40.0 (–CH<u>C</u>H$_2$CH–), 20.9 (Ar<u>C</u>H$_3$); HRMS (ESI) calculated for $C_{18}H_{19}N_3O_4$ [M + H]$^+$ 342.1454, found 342.1444.

**2″-Isoindolinyl-*N*-(4′-nitrophenyl)-2-pyrrolidinyl ketone (4n)**: yellow solid (0.76 g, 45%); m.p. 215–217°C; IR (neat) $v_{max}$ (cm$^{-1}$): 3075, 3060, 2953, 2854, 1654, 1581, 1512, 1479, 1288, 1195, 1106; $^{1}$H-NMR (400 MHz, CDCl$_3$) δ: 8.15 (1H, d, *J* = 8.0 Hz, *indolinyl* Ar<u>H</u>), 8.02 (2H, d, *J* = 9.0 Hz, Ar<u>H</u>), 7.23 (1H, d, *J* = 7.2 Hz, *indolinyl* Ar<u>H</u>), 7.15 (1H, t, *J* = 7.6 Hz, *indolinyl* Ar<u>H</u>), 7.05 (1H, t, *J* = 7.3 Hz, *indolinyl* Ar<u>H</u>), 6.37 (2H, d, *J* = 8.3 Hz, Ar<u>H</u>), 4.56 (1H, d, *J* = 7.9 Hz, –NC<u>H</u>CON′–), 4.35 (1H, dd, *J* = 17.3, 9.7 Hz, *indolinyl* –N′C<u>H</u>$_a$H$_b$–), 4.17 (1H, dd, *J* = 16.5, 9.8 Hz, *indolinyl* –N′C<u>H</u>$_{a'}$H$_b$–), 3.74 (1H, t, *J* = 7.2 Hz, *indolinyl* –N′CH$_a$<u>H</u>$_b$–), 3.54 (1H, dd, *J* = 16.2, 7.8 Hz, *indolinyl* –N′CH$_a$<u>H</u>$_{b'}$–), 3.33 (2H, dd, *J* = 16.4, 8.4 Hz, *prolyl* –C<u>H</u>$_2$N–), 2.45 (1H, dd, *J* = 17.9, 8.8 Hz, –CH$_2$C<u>H</u>$_a$H$_b$CH–), 2.32–2.11 (3H, m, –C<u>H</u>$_2$CH$_a$<u>H</u>$_b$CH–); $^{13}$C-NMR (150 MHz, CDCl$_3$) δ: 169.2 (–<u>C</u>ON′–), 151.0; 142.7; 137.4; 130.8; 127.7; 126.3; 124.7; 124.4; 117.3; 110.9 (<u>Ar</u>H), 61.0 (–<u>C</u>HCON′–), 49.0; 47.6 (*indolinyl* –N′<u>C</u>H$_2$–), 30.4 (*prolyl* –<u>C</u>H$_2$N–), 28.4 (–CH$_2$<u>C</u>H$_2$CH–), 23.1 (–<u>C</u>H$_2$CH$_2$CH-); HRMS (ESI) calculated for $C_{19}H_{19}N_3O_3$ [M + Na]$^+$ 360.1324, found 360.1323.

***N*-(4′-nitrophenyl)-*N*′,*N*′-dipropyl-ʟ-prolinamide (4o)**: yellow solid (0.77 g, 48%); m.p. 136–138°C; IR (neat) $v_{max}$ (cm$^{-1}$): 3025, 2961, 2922, 2873, 1637, 1597, 1519, 1488, 1300, 1230, 1109; $^{1}$H-NMR (600 MHz, CDCl$_3$) δ: 8.06 (2H, d, *J* = 9.4 Hz, Ar<u>H</u>), 6.35 (2H, d, *J* = 8.5 Hz, Ar<u>H</u>), 4.57 (1H, dd, *J* = 8.5, 2.5 Hz, –NC<u>H</u>CON′–), 3.68 (1H, td, *J* = 9.0, 3.7 Hz, *prolyl* –C<u>H</u>$_a$H$_b$N–), 3.51 (1H, dd, *J* = 16.9, 7.7 Hz, *prolyl* –CH$_a$<u>H</u>$_b$N–), 3.43–3.33 (2H, m, –N′C<u>H</u>$_2$CH$_2$CH$_3$), 3.31–3.21 (2H, m, –N′C<u>H</u>$_2$CH$_2$CH$_3$), 2.38 (1H, ddd, *J* = 17.8, 8.2, 5.8 Hz, –CH$_2$C<u>H</u>$_a$H$_b$CH–), 2.25–2.17 (1H, m, –CH$_2$CH$_a$<u>H</u>$_b$CH–), 2.11–2.03 (2H, m, –C<u>H</u>$_2$CH$_a$H$_b$CH–), 1.75 (2H, dd, *J* = 14.9, 7.4 Hz, –N′CH$_2$C<u>H</u>$_2$CH$_3$), 1.55 (2H, dd, *J* = 15.0, 7.5 Hz, –N′CH$_2$C<u>H</u>$_2$CH$_3$), 1.02 (3H, t, *J* = 7.4 Hz, –N′CH$_2$CH$_2$C<u>H</u>$_3$), 0.86 (3H, t, *J* = 7.4 Hz, –N′CH$_2$CH$_2$C<u>H</u>$_3$); $^{13}$C-NMR (150 MHz, CDCl$_3$) δ: 170.7 (–<u>C</u>ON′–), 151.3; 137.1; 126.2; 110.8 (<u>Ar</u>H), 59.1 (–<u>C</u>HCON′–), 49.2; 49.0 (–N′<u>C</u>H$_2$CH$_2$CH$_3$), 47.9 (–<u>C</u>H$_2$N–), 31.1 (–CH$_2$<u>C</u>H$_2$CH–), 23.4 (–<u>C</u>H$_2$CH$_2$CH–), 22.5; 20.8 (–N′CH$_2$<u>C</u>H$_2$CH$_3$), 11.4; 11.3 (–N′CH$_2$CH$_2$<u>C</u>H$_3$); HRMS (ESI) calculated for $C_{17}H_{25}N_3O_3$ [M + H]$^+$ 320.1974, found 320.1998.

***N*-(4′-nitrophenyl)-*N*′-(2″-pyridinyl)-ʟ-prolinamide (4p)**: yellow solid (0.72 g, 46%); m.p. 180–182°C; IR (neat) $v_{max}$ (cm$^{-1}$): 3290, 3080, 2980, 2876, 1708, 1595, 1515, 1475, 1333, 1290, 1197, 1110; $^{1}$H-NMR (600 MHz, CDCl$_3$) δ: 8.50 (1H, d, *J* = 12.6 Hz, –CON<u>H</u>), 8.25 (1H, d, *J* = 8.2 Hz, *pyridinyl* –C<u>H</u>–), 8.20 (1H, s, *pyridinyl* –C<u>H</u>–), 8.15–8.08 (2H, m, Ar<u>H</u>), 7.74–7.69 (1H, m, *pyridinyl* –C<u>H</u>–), 7.05 (1H, dd, *J* = 7.3, 3.5 Hz, *pyridinyl* –C<u>H</u>–), 6.65–6.61 (2H, m, Ar<u>H</u>), 4.30 (1H, dd, *J* = 8.6, 2.8 Hz, –NC<u>H</u>CON′–), 3.85 (1H, dd, *J* = 7.4, 5.2 Hz, –C<u>H</u>$_a$H$_b$N–), 3.44 (1H, q, *J* = 8.8 Hz, –CH$_a$<u>H</u>$_b$N–), 2.48–2.36 (2H, m, –CH$_2$C<u>H</u>$_2$CH–), 2.15 (2H, dt, *J* = 13.6, 6.9 Hz, –C<u>H</u>$_2$CH$_2$CH–); $^{13}$C-NMR (150 MHz, CDCl$_3$) δ: 171.1 (–<u>C</u>ON′–), 151.6 (<u>Ar</u>H), 150.4; 147.8; 139.1 (*pyr*-<u>Ar</u>H), 138.5; 126.1 (<u>Ar</u>H), 120.4; 114.0 (*pyr*-<u>Ar</u>H), 112.1 (<u>Ar</u>H), 64.6 (–<u>C</u>HCON′–), 49.8 (–<u>C</u>H$_2$N–), 31.7 (–CH$_2$<u>C</u>H$_2$CH–), 24.0 (–<u>C</u>H$_2$CH$_2$CH–); HRMS (ESI) calculated for $C_{16}H_{16}N_4O_3$ [M + H]$^+$ 313.1301, found 313.1312.

***N*′-benzyl-*N*-(4′-nitrophenyl)-ʟ-prolinamide (4q)**: yellow solid (0.89 g, 55%); m.p. 178–179°C; IR (neat) $v_{max}$ (cm$^{-1}$): 3292, 3085, 2979, 2952, 2877, 1707, 1648, 1513, 1479, 1396, 1293, 1198, 1115; $^{1}$H-NMR (400 MHz, CDCl$_3$) δ: 8.01 (2H, d, *J* = 8.9 Hz, Ar<u>H</u>), 7.25 (1H, s, –CON<u>H</u>), 7.24–7.09 (4H, m, Ar<u>H</u>), 6.51 (3H, m, Ar<u>H</u>), 4.50–4.35 (2H, m, ArC<u>H</u>$_2$–), 4.21 (1H, t, *J* = 5.4 Hz, –NC<u>H</u>CON′–), 3.68 (1H, t, *J* = 7.9 Hz, –C<u>H</u>$_a$H$_b$N–), 3.35 (1H, dd, *J* = 16.4, 9.5 Hz, –CH$_a$<u>H</u>$_b$N–), 2.43–2.29 (2H, m, –CH$_2$C<u>H</u>$_2$CH–), 2.05 (2H, dt, *J* = 20.3, 11.7 Hz, –C<u>H</u>$_2$CH$_2$CH–); $^{13}$C-NMR (150 MHz, CDCl$_3$) δ: 171.9 (–<u>C</u>ON′–), 151.8; 138.6; 137.8; 128.7; 127.6; 127.4; 126.0; 111.9 (<u>Ar</u>H), 64.0 (–<u>C</u>HCON′–), 49.5 (–<u>C</u>H$_2$N–), 43.4 (Ar<u>C</u>H$_2$–), 31.5 (–CH$_2$<u>C</u>H$_2$CH–), 24.0 (–<u>C</u>H$_2$CH$_2$CH–); HRMS (ESI) calculated for $C_{18}H_{19}N_3O_3$ [M + H]$^+$ 326.1505, found 326.1510.

***N*′-(2″-cyanophenyl)-*N*-(4′-nitrophenyl)-ʟ-prolinamide (4r)**: yellow solid (0.59 g, 35%); m.p. 211–213°C; IR (neat) $v_{max}$ (cm$^{-1}$): 3257, 3060, 2950, 2930, 2867, 2226, 1682, 1596, 1484, 1450, 1390, 1312, 1166, 1112; $^{1}$H-NMR (400 MHz, CDCl$_3$) δ: 8.56 (1H, s, –CON<u>H</u>), 8.35 (1H, d, *J* = 8.4 Hz, Ar<u>H</u>), 8.14 (2H, d, *J* = 9.1 Hz, Ar<u>H</u>), 7.60 (1H, t, *J* = 7.8 Hz, Ar<u>H</u>), 7.52 (1H, d, *J* = 7.6 Hz, Ar<u>H</u>), 7.19 (1H, t, *J* = 7.6 Hz, Ar<u>H</u>), 6.69 (2H, d, *J* = 9.1 Hz, Ar<u>H</u>), 4.36 (1H, dd, *J* = 8.0, 3.0 Hz, –NC<u>H</u>CON′–), 4.00–3.89 (1H, m, –C<u>H</u>$_a$H$_b$N–), 3.49 (1H, dd, *J* = 17.4, 8.9 Hz, –CH$_a$<u>H</u>$_b$N–), 2.53–2.38 (2H, m, –CH$_2$C<u>H</u>$_2$CH–), 2.23 (2H, dt, *J* = 11.2, 5.8 Hz,

–C$\underline{H}_2$CH$_2$CH–); $^{13}$C-NMR (150 MHz, CDCl$_3$) δ: 171.0 (–$\underline{C}$ON'–), 151.3; 139.5; 139.4; 134.2; 132.0; 126.2; 124.8; 121.0 (Ar$\underline{H}$), 115.8 (–$\underline{C}$N), 112.3; 102.8 (Ar$\underline{H}$), 64.6 (–$\underline{C}$HCON'–), 49.7 (–$\underline{C}$H$_2$N–), 31.6 (–CH$_2\underline{C}$H$_2$CH–), 24.0 (–$\underline{C}$H$_2$CH$_2$CH–); HRMS (ESI) calculated for C$_{18}$H$_{16}$N$_4$O$_3$ [M + H]$^+$ 337.1301, found 337.1313.

$N'$-methyl-$N$-(4'-nitrophenyl)-$N'$-phenyl-L-prolinamide (4s): yellow solid (0.57 g, 33%); m.p. 186–188°C; IR (neat) $v_{max}$ (cm$^{-1}$): 3055, 3042, 2952, 2920, 2868, 1656, 1593, 1483, 1378, 1303, 1193, 1111; $^1$H-NMR (400 MHz, CDCl$_3$) δ: 8.09 (2H, d, $J$ = 8.8 Hz, Ar$\underline{H}$), 7.53 (2H, t, $J$ = 7.3 Hz, Ar$\underline{H}$), 7.44 (1H, t, $J$ = 7.2 Hz, Ar$\underline{H}$), 7.34 (2H, d, $J$ = 7.5 Hz, Ar$\underline{H}$), 6.35 (2H, d, $J$ = 8.7 Hz, Ar$\underline{H}$), 4.24 (1H, d, $J$ = 5.9 Hz, –NC$\underline{H}$CON'–), 3.72–3.63 (1H, m, –C$\underline{H}_a$H$_b$N–), 3.44 (1H, dd, $J$ = 15.8, 8.1 Hz, –CH$_a\underline{H}_b$N–), 3.30 (3H, s, –N'C$\underline{H}_3$), 2.34–1.94 (4H, m, –C$\underline{H}_2$C$\underline{H}_2$CH–); $^{13}$C-NMR (150 MHz, CDCl$_3$) δ: 171.8 (–$\underline{C}$ON'–), 151.0; 142.8; 137.3; 130.3; 128.6; 127.2; 126.2; 110.8 (Ar$\underline{H}$), 59.1 (–$\underline{C}$HCON'–), 49.1 (–$\underline{C}$H$_2$N–), 37.9 (–N'$\underline{C}$H$_3$), 31.5 (–CH$_2\underline{C}$H$_2$CH–), 23.6 (–$\underline{C}$H$_2$CH$_2$CH–); HRMS (ESI) calculated for C$_{18}$H$_{19}$N$_3$O$_3$ [M + Na]$^+$ 348.1324, found 348.1303.

$N'$-(2''-(hydroxymethyl)phenyl)-$N$-(4'-nitrophenyl)-L-prolinamide (4t): yellow solid (0.90 g, 53%); m.p. 179–180°C; IR (neat) $v_{max}$ (cm$^{-1}$): 3248, 3080, 2980, 2952, 2869, 1661, 1584, 1537, 1512, 1482, 1312, 1181, 1112; $^1$H-NMR (600 MHz, CDCl$_3$) δ: 9.52 (1H, s, –CON$\underline{H}$), 8.15 (1H, d, $J$ = 8.2 Hz, Ar$\underline{H}$), 8.10 (2H, d, $J$ = 9.1 Hz, Ar$\underline{H}$), 7.34–7.29 (1H, m, Ar$\underline{H}$), 7.09–7.02 (2H, m, Ar$\underline{H}$), 6.62 (2H, d, $J$ = 9.2 Hz, Ar$\underline{H}$), 4.47 (1H, d, $J$ = 12.6 Hz, –NC$\underline{H}$CON'–), 4.30 (2H, dd, $J$ = 11.9, 6.1 Hz, –C$\underline{H}_2$OH), 3.87–3.82 (1H, m, –C$\underline{H}_a$H$_b$N–), 3.47 (1H, dd, $J$ = 16.8, 9.6 Hz, –CH$_a\underline{H}_b$N–), 2.40 (2H, dd, $J$ = 9.6, 5.7 Hz, –CH$_2\underline{C}$H$_2$CH–), 2.15 (2H, dt, $J$ = 12.3, 5.7 Hz, –C$\underline{H}_2$CH$_2$CH–), 1.68 (1H, s, –CH$_2$O$\underline{H}$); $^{13}$C-NMR (150 MHz, CDCl$_3$) δ: 170.5 (–$\underline{C}$ON'–), 151.3; 138.5; 136.9; 129.2; 128.9; 128.6; 126.1; 124.5; 121.6; 111.8 (Ar$\underline{H}$), 64.6 (–$\underline{C}$HCON'–), 64.4 (–$\underline{C}$H$_2$OH), 49.2 (–$\underline{C}$H$_2$N–), 31.5 (–CH$_2\underline{C}$H$_2$CH–), 23.9 (–$\underline{C}$H$_2$CH$_2$CH–); HRMS (ESI) calculated for C$_{18}$H$_{19}$N$_3$O$_4$ [M + H]$^+$ 342.1454, found 342.1471.

$N,N'$-bis(4'-nitrophenyl)-L-prolinamide (4u): yellow solid (0.78 g, 44%); m.p. 137–139°C; IR (neat) $v_{max}$ (cm$^{-1}$): 3318, 3085, 2921, 2869, 1687, 1595, 1499, 1292, 1158, 1108; $^1$H-NMR (600 MHz, DMSO-$d_6$) δ: 9.95 (1H, s, –CON$\underline{H}$), 7.33 (2H, d, $J$ = 9.2 Hz, Ar$\underline{H}$), 7.18 (2H, d, $J$ = 9.4 Hz, Ar$\underline{H}$), 6.97 (2H, d, $J$ = 9.2 Hz, Ar$\underline{H}$), 5.72 (2H, d, $J$ = 8.1 Hz, Ar$\underline{H}$), 3.68 (1H, dd, $J$ = 8.8, 2.1 Hz, –NC$\underline{H}$CON'–), 2.81–2.76 (1H, m, –C$\underline{H}_a$H$_b$N–), 2.59 (1H, dd, $J$ = 17.6, 8.1 Hz, –CH$_a\underline{H}_b$N–), 1.57–1.49 (1H, m, –CH$_2$C$\underline{H}_a$H$_b$CH–), 1.30 (1H, dt, $J$ = 6.9, 5.7 Hz, –CH$_2$CH$_a\underline{H}_b$CH–), 1.21–1.15 (2H, m, –C$\underline{H}_2$CH$_2$CH–); $^{13}$C-NMR (150 MHz, DMSO-$d_6$) δ: 171.8 (–$\underline{C}$ON'–), 151.8; 145.2; 142.9; 136.8; 126.4; 125.3; 119.7; 111.8 (Ar$\underline{H}$), 62.6 (–$\underline{C}$HCON'–), 49.3 (–$\underline{C}$H$_2$N–), 31.7 (–CH$_2\underline{C}$H$_2$CH–), 23.7 (–$\underline{C}$H$_2$CH$_2$CH–); HRMS (ESI) calculated for C$_{17}$H$_{16}$N$_4$O$_5$ [M + Na]$^+$ 379.1018, found 379.1007.

$N'$-isopropyl-$N$-(4'-nitrophenyl)-L-prolinamide (4v): yellow solid (0.69 g, 50%); m.p. 231–232°C; IR (neat) $v_{max}$ (cm$^{-1}$): 3275, 3081, 2972, 2870, 1647, 1597, 1548, 151, 1483, 1303, 1238, 1188, 1111; $^1$H-NMR (400 MHz, CDCl$_3$) δ: 8.12 (2H, d, $J$ = 9.0 Hz, Ar$\underline{H}$), 6.56 (2H, d, $J$ = 9.0 Hz, Ar$\underline{H}$), 5.84 (1H, d, $J$ = 7.6 Hz, –CON$\underline{H}$), 4.16–4.04 (2H, m, –NC$\underline{H}$CON'N'–; –N'C$\underline{H}$(CH$_3$)$_2$), 3.72 (1H, t, $J$ = 8.1 Hz, –C$\underline{H}_a$H$_b$N–), 3.38 (1H, dd, $J$ = 16.5, 9.6 Hz, –CH$_a\underline{H}_b$N–), 2.31 (2H, dd, $J$ = 13.1, 9.3 Hz, –CH$_2\underline{C}$H$_2$CH–), 2.17–1.96 (2H, m, –C$\underline{H}_2$CH$_2$CH–), 1.07 (6H, dd, $J$ = 28.0, 6.5 Hz, –N'CH(C$\underline{H}_3$)$_2$); $^{13}$C-NMR (150 MHz, CDCl$_3$) δ: 170.8 (–$\underline{C}$ON'–), 151.6; 138.7; 126.1; 111.8 (Ar$\underline{H}$), 64.1 (–$\underline{C}$HCON'–), 49.6 (–$\underline{C}$H$_2$N–), 41.5 (–N'$\underline{C}$H(CH$_3$)$_2$), 31.4 (–CH$_2\underline{C}$H$_2$CH–), 23.9 (–$\underline{C}$H$_2$CH$_2$CH–), 22.7; 22.5 (–N'CH($\underline{C}$H$_3$)$_2$); HRMS (ESI) calculated for C$_{14}$H$_{19}$N$_3$O$_3$ [M + H]$^+$ 278.1505, found 278.1494.

$N',N'$-dicyclohexyl-$N$-(4'-nitrophenyl)-L-prolinamide (4w): yellow solid (0.60 g, 30%); m.p. 140–142°C; IR (neat) $v_{max}$ (cm$^{-1}$): 3031, 2926, 2851, 1648, 1596, 1512, 1484, 1289, 1234, 1182, 1108; $^1$H-NMR (600 MHz, CDCl$_3$) δ: 8.07 (2H, d, $J$ = 9.3 Hz, Ar$\underline{H}$), 6.36 (2H, d, $J$ = 4.9 Hz, Ar$\underline{H}$), 4.52 (1H, dd, $J$ = 8.6, 2.7 Hz, –NC$\underline{H}$CON'–), 3.66 (1H, td, $J$ = 8.9, 4.0 Hz, cyclohexyl –C$\underline{H}$–), 3.61–3.51 (2H, m, cyclohexyl –C$\underline{H}$–; –C$\underline{H}_a$H$_b$N–), 2.39–2.32 (2H, m, –CH$_a\underline{H}_b$N–; –CH$_2$C$\underline{H}_a$H$_b$CH–), 2.24–2.16 (1H, m, –CH$_2$CH$_a\underline{H}_b$CH–), 2.11–1.98 (2H, m, cyclohexyl –C$\underline{H}_2$–), 1.91 (3H, m, –C$\underline{H}_2$CH$_a$H$_b$CH–; cyclohexyl –C$\underline{H}_a$H$_b$–), 1.74 (4H, m, cyclohexyl –CH$_a\underline{H}_b$–; –C$\underline{H}_2$–), 1.51–1.07 (13H, m, cyclohexyl –C$\underline{H}_2$–); $^{13}$C-NMR (150 MHz, CDCl$_3$) δ: 169.5 (–$\underline{C}$ON'–), 151.3; 137.1; 126.1; 110.8 (Ar$\underline{H}$), 60.5 (–$\underline{C}$HCON'–), 57.6; 56.7 (cyclohexyl –$\underline{C}$H–), 49.0 (–$\underline{C}$H$_2$N–), 31.6; 31.1 (cyclohexyl –$\underline{C}$H$_2$–), 30.8 (prolyl –CH$_2\underline{C}$H$_2$CH–), 30.0; 29.6 (cyclohexyl –$\underline{C}$H$_2$–), 26.50 (prolyl –$\underline{C}$H$_2$CH$_2$CH–), 26.48; 26.02; 25.97; 25.2; 25.1; 23.4 (cyclohexyl –$\underline{C}$H$_2$–); HRMS (ESI) calculated for C$_{23}$H$_{33}$N$_3$O$_3$ [M + H]$^+$ 400.2600, found 400.2604.

# 6. Biology

## 6.1. Materials

3-(4,5-Dimethylthiazol-2-yl)-2,5-diphenyltetrazolium bromide (MTT), 5-fluorourasil and dimethyl sulfoxide (DMSO) were purchased from Sigma and Merck, respectively, whereas fetal bovine serum

(FBS) and phosphate-buffered saline (PBS) were both obtained from Gibco. Dulbecco's Modified Eagle Medium (DMEM) was purchased from Hyclone.

## 6.2. Cell lines and culture

Four human cancer cell lines: gastric carcinoma (SGC7901), colon carcinoma (HCT-116), liver carcinoma (HepG2) and lung carcinoma (A549) cell lines were used in this study. All the cell lines were grown as adherent monolayers in flasks with DMEM cultured media supplemented with 10% FBS and 1% streptomycin/penicillin in a humidified incubator with 5% $CO_2$ at 37°C.

## 6.3. MTT assay

The cytotoxicies of L-prolinamides **4a–4w** were estimated against human gastric (SGC7901), colon (HCT-116), liver (HepG2) and lung (A549) cancer cell lines using the MTT assay. The cells were seeded evenly in 96-well plates with a cell density of 3000 cells well$^{-1}$ with FBS and incubated overnight for attachment. After 24 h, 100 µl (per well) of different concentrations of L-prolinamides **4a–4w** in serial 10-fold dilutions (1 µM, 10 µM and 100 µM) were added in triplicates into the wells and incubated at 5% $CO_2$ at 37°C. Stock solutions of L-prolinamides **4a–4w** were prepared in DMSO; followed by dilution in buffer so that the final concentration of DMSO in the culture media was 1%. After 72 h of incubation, the culture medium was carefully removed and MTT solution (100 µl) was added into each well and incubated at 37°C for 2 h, to allow the viable cells to bio-transform the yellow-coloured MTT into dark-blue formazan crystals. The MTT solution was then aspirated and DMSO (150 µl) was added into each well and incubated at 37°C at 800 r.p.m. for 10 min, to dissolve the formazan crystals. The optical density (absorbance) of each sample was measured at 490 nm using the M3 SpectraMax microplate reader (Molecular Devices).

The percentage cell viability and inhibition were calculated using equations (6.1) and (6.2), and were used as a measure of the compounds' inhibition potencies on the cancer cell lines:

$$\text{cell viability (\%)} = \frac{\text{OD}_{\text{sample}}}{\text{OD}_{\text{control}}} \times 100\% \tag{6.1}$$

and

$$\% \text{ inhibition} = 100\% - \text{cell viability (\%)}. \tag{6.2}$$

Data accessibility. NMR spectroscopic data for the synthesized compounds are included as the electronic supplementary material. Data are also deposited in the Dryad Digital Repository [34].

Authors' contributions. A.O. participated in the design of the study and carried out the synthesis; J.I. participated in the design of the study and data analyses, drafted and submitted the manuscript; X.B. coordinated the benchwork and reviewed the draft manuscript; O.A. participated in the design of the study and reviewed the draft manuscript; J.K. carried out the assays; C.G. coordinated the assays and reviewed the draft manuscript; O.F. led the conception and design of the study, coordinated the study and reviewed drafts. All authors approved the manuscript for publication and agree to be held accountable for the work performed therein.

Competing interests. There are no conflicts of interest to declare.

Funding. This work was funded, in part, by a University of Lagos Central Research Committee grant (CRC no. 2015/25).

Acknowledgement. The authors are grateful for the support of this research by Soochow University, P. R. China, University of Lagos, Nigeria and Tai Solarin University of Education, Nigeria.

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
