## [Reviewer comments · Royal Society Open Science]

Review History

RSOS-200906.R0 (Original submission)

Review form: Reviewer 1

Is the manuscript scientifically sound in its present form?

Yes

Are the interpretations and conclusions justified by the results?

Yes

Is the language acceptable?

Yes

Do you have any ethical concerns with this paper?

No

Have you any concerns about statistical analyses in this paper?

No

Recommendation?

Accept with minor revision (please list in comments)

Comments to the Author(s)

In this manuscript Profs. Bao and FAMILONI describe the synthesis and evaluation of a series of N-arylated prolinamides as in vitro anticancer agents. The work is scientifically sound, and identified antiproliferative agents that could lead to further refinement and follow-up studies. I am therefore recommending publication of the present manuscript in Royal Society Open Science, after the following comments have been addressed by the authors:

1. Please include additional information on the rationale of your structural design. The tested compounds have novel structures, and are quite different from the structures in figure 1. (those are either N-acyl or free amino versus N-aryl substitutions in the present study), and therefore there is no indication as to why the authors thought these molecules might have applications as anticancer agents.
2. Related to the first point, please indicate what was the criteria to select the cell lines tested. Is any information about the potential mechanism of action or molecular target available? If so, please include it in your discussion.
3. The activity observed ranges from modest to good, but the potential selectivity could be equally important or even more relevant. Please state whether any selectivity study using cell line models for "healthy cells" was conducted, or state whether that would be part of a planned follow-up study.
4. For analog 4m, indicate in the structure the stereochemistry of the hydroxyl group at position 4 of the proline.

Review form: Reviewer 2

Is the manuscript scientifically sound in its present form?

No

Are the interpretations and conclusions justified by the results?

Yes

Is the language acceptable?

Yes

Do you have any ethical concerns with this paper?

No

Have you any concerns about statistical analyses in this paper?

No

Recommendation?

Major revision is needed (please make suggestions in comments)

Comments to the Author(s)

The authors describe the synthesis and anticancer activity of a series of N-substituted(4'-nitrophenyl)-L-prolinamides. The compounds are interesting with a number (notably 4a and 4u) possessing greater activity than 5-FU control). However, a number of issues need attention before I can support publication:

Major issues:

No information on the purity of the compounds is provided. The compounds need HPLC or elemental analysis determined purity levels to ensure that an impurity is not responsible for the assigned activity.

P4L16 - 'peptides are typically favored over small organic molecules in drug development' This is not true (at least in the way the authors are describing) peptide or peptidomimetic compounds represent a very small % of approved drugs. Small molecules are preferred for the reasons discussed by the authors.

The authors screen their compounds at 1, 10 and 100 uM concentrations but why not add another concentration and determine an IC50 value? An IC50 is much more robust and allows better comparison between derivatives.

The discussion of SAR wherein longer chains provide more active compounds is a common observation in medicinal chemistry. The greater lipophilicity afforded by the longer chains enable greater cell uptake. See Wang et. al. ChemMedChem. 2017, 12, 1033. Consider adding this reference.

No selectivity data in a non-cancerous cell line is presented, despite the introduction discussing non-selective toxicity as a real problem. This should be determined at least for the most active compounds. This is especially important with compound 4u which is toxic against all the cells tested. Is this just a toxic compound?

P12L21. '4k and 4m suspected of encouraging proliferation' This seems unlikely, is this not just general growth? A negative control in the experiment would determine levels of growth in vehicle alone.

P12L30; Sulphonamides alone are not a pharmacophore, it is their substituents so this sentence is incorrect.

P12:35; 4l has no inhibition activity or it just wasn't determined as shown in Table 2? It is unlikely that the physical state will be relevant to activity.

Minor issues:

The abstract is very long and switches between discussing cell inhibition % and cell survival % which are not the same thing.

The prolinamide structure in the nucleoside compounds in figure 1 is a little misleading as these are ligands of a protide structure that will be lost in the cell, it has no contribution to the pharmacophore.

Scheme 1 shows 3a-d but the text only refers to 3a-c.

A line to separate cell types in table 2 would be useful for clarity of presentation.

Decision letter (RSOS-200906.R0)

Dear Dr Izunobi:

Title: Synthesis and in vitro Anticancer Activities of Substituted N-(4'-Nitrophenyl)-L-prolinamides
Manuscript ID: RSOS-200906

The editor assigned to your manuscript has now received comments from reviewers. We would like you to revise your paper in accordance with the referee and Subject Editor suggestions which can be found below (not including confidential reports to the Editor). Please note this decision does not guarantee eventual acceptance.

Please submit your revised paper before 02-Aug-2020. Please note that the revision deadline will expire at 00.00am on this date. If we do not hear from you within this time then it will be assumed that the paper has been withdrawn. In exceptional circumstances, extensions may be possible if agreed with the Editorial Office in advance. We do not allow multiple rounds of revision so we urge you to make every effort to fully address all of the comments at this stage. If deemed necessary by the Editors, your manuscript will be sent back to one or more of the original reviewers for assessment. If the original reviewers are not available we may invite new reviewers.

On behalf of the Subject Editor Professor Anthony Stace and the Associate Editor Dr Andrew Harned.

RSC Associate Editor:

Comments to the Author:

The reviewers have expressed some enthusiasm for the work presented in this manuscript. I agree that while the reported activity is not spectacular, there is enough activity present that others may find inspiration in these compounds/motifs. Thus, the work is suitable for this journal.

However, the reviewers raise a number of valid concerns that should be addressed by the authors. In particular, addressing the concern over compound is a necessity. Also, I am inclined to agree that some toxicity data in healthy cells for the most active compounds is required.

I ask the authors to carefully consider the attached reviewer comments and submit a revised manuscript. Given, the present global circumstances, I ask the authors to do their best at addressing these concerns. There may be some concerns that require additional experimentation. If this is not possible, please justify why we should consider a revised manuscript without these experiments.

RSC Subject Editor:

Comments to the Author:

(There are no comments.)

Reviewers' Comments to Author:

Reviewer: 1

Comments to the Author(s)

In this manuscript Profs. Bao and Familoni describe the synthesis and evaluation of a series of N-arylated prolinamides as in vitro anticancer agents. The work is scientifically sound, and identified antiproliferative agents that could lead to further refinement and follow-up studies. I am therefore recommending publication of the present manuscript in Royal Society Open Science, after the following comments have been addressed by the authors:

1. Please include additional information on the rationale of your structural design. The tested compounds have novel structures, and are quite different from the structures in figure 1. (those are either N-acyl or free amino versus N-aryl substitutions in the present study), and therefore there is no indication as to why the authors thought these molecules might have applications as anticancer agents.
2. Related to the first point, please indicate what was the criteria to select the cell lines tested. Is any information about the potential mechanism of action or molecular target available? If so, please include it in your discussion.
3. The activity observed ranges from modest to good, but the potential selectivity could be equally important or even more relevant. Please state whether any selectivity study using cell line models for "healthy cells" was conducted, or state whether that would be part of a planned follow-up study.
4. For analog 4m, indicate in the structure the stereochemistry of the hydroxyl group at position 4 of the proline.

Reviewer: 2

Comments to the Author(s)

The authors describe the synthesis and anticancer activity of a series of N-substituted(4'-nitrophenyl)-L-prolinamides. The compounds are interesting with a number (notably 4a and 4u) possessing greater activity than 5-FU control). However, a number of issues need attention before I can support publication:

Major issues:

No information on the purity of the compounds is provided. The compounds need HPLC or elemental analysis determined purity levels to ensure that an impurity is not responsible for the assigned activity.

P4L16 - 'peptides are typically favored over small organic molecules in drug development' This is not true (at least in the way the authors are describing) peptide or peptidomimetic compounds represent a very small % of approved drugs. Small molecules are preferred for the reasons discussed by the authors.

The authors screen their compounds at 1, 10 and 100 uM concentrations but why not add another concentration and determine an IC50 value? An IC50 is much more robust and allows better comparison between derivatives.

The discussion of SAR wherein longer chains provide more active compounds is a common observation in medicinal chemistry. The greater lipophilicity afforded by the longer chains enable greater cell uptake. See Wang et. al. ChemMedChem. 2017, 12, 1033. Consider adding this reference.

No selectivity data in a non-cancerous cell line is presented, despite the introduction discussing non-selective toxicity as a real problem. This should be determined at least for the most active compounds. This is especially important with compound 4u which is toxic against all the cells tested. Is this just a toxic compound?

P12L21. '4k and 4m suspected of encouraging proliferation' This seems unlikely, is this not just general growth? A negative control in the experiment would determine levels of growth in vehicle alone.

P12L30; Sulphonamides alone are not a pharmacophore, it is their substituents so this sentence is incorrect.

P12:35; 4l has no inhibition activity or it just wasn't determined as shown in Table 2? It is unlikely that the physical state will be relevant to activity.

Minor issues:

The abstract is very long and switches between discussing cell inhibition % and cell survival % which are not the same thing.

The prolinamide structure in the nucleoside compounds in figure 1 is a little misleading as these are ligands of a protide structure that will be lost in the cell, it has no contribution to the pharmacophore.

Scheme 1 shows 3a-d but the text only refers to 3a-c.

A line to separate cell types in table 2 would be useful for clarity of presentation.

Author's Response to Decision Letter for (RSOS-200906.R0)

See Appendix A.

Decision letter (RSOS-200906.R1)

Dear Dr Izunobi:

Title: Synthesis and *in vitro* Anticancer Activities of Substituted *N*-(4'-Nitrophenyl)-*L*-prolinamides

Manuscript ID: RSOS-200906.R1

It is a pleasure to accept your manuscript in its current form for publication in Royal Society Open Science. The chemistry content of Royal Society Open Science is published in collaboration with the Royal Society of Chemistry.

On behalf of the Subject Editor Professor Anthony Stace and the Associate Editor Dr Andrew Harned.

RSC Associate Editor

Comments to the Author:

The authors have addressed most of the concerns raised by the referees, and have plans to address the others in subsequent work. I recommend publication at this time.

Reviewer(s)' Comments to Author:

Appendix A

Response to Referees

Journal Name: Royal Society Open Science

Journal Code: RSOS

Online ISSN: [2054-5703](https://doi.org/10.1093/rsof/rzab000)

Journal Admin Email: openscience@royalsociety.org

Journal Editor: Dr Ellis Wilde

Journal Editor Email: chemistryopenscience@rsc.org

MS Reference Number: RSOS-200906

Article Status: SUBMITTED

MS Dryad ID: RSOS-200906

MS Title: Synthesis and in vitro Anticancer Activities of Substituted *N*-(4'-Nitrophenyl)-L-prolinamides

MS Authors: Osinubi, Adejoke; Izunobi, Josephat; Bao, Xiaoguang ; Asekun, Olayinka ; Kong, Jiehong; Gui, Chunshan; Familoni, Oluwole

Contact Author: Josephat Izunobi

Contact Author Email: jizunobi@gmail.com

RSC Associate Editor's Comment

The reviewers have expressed some enthusiasm for the work presented in this manuscript. I agree that while the reported activity is not spectacular, there is enough activity present that others may find inspiration in these compounds/motifs. Thus, the work is suitable for this journal.

Authors' Comment

Thank you for your comments.

RSC Associate Editor's Comment

However, the reviewers raise a number of valid concerns that should be addressed by the authors. In particular, addressing the concern over compound is a necessity. Also, I am inclined to agree that some toxicity data in healthy cells for the most active compounds is required.

Authors' Comment

The concerns are valid. The authors' intension was to first identify leads, zero in on them and then carry out more exhaustive assays, to minimise cost, amongst other considerations. Toxicity assays on healthy analogous human cells will be carried out in the future.

RSC Associate Editor's Comment

I ask the authors to carefully consider the attached reviewer comments and submit a revised manuscript. Given, the present global circumstances, I ask the authors to do their best at addressing these concerns. There may be some concerns that require additional experimentation. If this is not possible, please justify why we should consider a revised manuscript without these experiments.

Authors' Comment

The authors have carefully considered and responded (*vide infra*) to the Reviewers' comments, to the best of their abilities, and hereby submit a revised manuscript. The suggested additional experimentation, though unattainable in the present clime, do not – in the authors' opinion – diminish the results presented herein as some of these experiments are planned for future work. This submission, though propelled by our current global circumstances, will encourage salient discussions, in this area, till we are all able to return to the laboratory.

Reviewer 1's Comment

In this manuscript Profs. Bao and Familoni describe the synthesis and evaluation of a series of *N*-arylated prolinamides as *in vitro* anticancer agents. The work is scientifically sound, and identified antiproliferative agents that could lead to further refinement and follow-up studies.

I am therefore recommending publication of the present manuscript in Royal Society Open Science, after the following comments have been addressed by the authors:

Authors' Comment

The authors thank this Reviewer.

Reviewer 1's Comment

1. Please include additional information on the rationale of your structural design. The tested compounds have novel structures, and are quite different from the structures in figure 1. (those are either *N*-acyl or free amino versus *N*-aryl substitutions in the present study), and therefore there is no indication as to why the authors thought these molecules might have applications as anticancer agents.

Authors' Comment

- The authors have inserted a new paragraph on p.4, line 26ff:

“The *N*-functionalisation of amino acids with aryl compounds has great utility in organic syntheses as well as drug discovery and pharmaceuticals.^{9,10} *N*-aryl amino acids are, therefore, employed as inexpensive chiral building blocks and are crucial motifs in many systems of physiological importance.¹¹ *N*-arylated amino acid derivatives can also be incorporated into peptides and proteins to propel the development of new methods for the study of protein structures and functions, amongst others.⁷ Pertinently, *N*-arylation reactions have been exploited in the introduction of diversity into bioactive molecules and synthesis of anti-cancer agents with improved potencies.¹⁰”

- The authors also inserted a sentence on p.5, line 9:

“Additionally, the *N*-arylation of prolinamides, if synergistic, should bode well for the resulting substrates' antineoplastic potencies.”

Reviewer 1's Comment

2. Related to the first point, please indicate what was the criteria to select the cell lines tested. Is any information about the potential mechanism of action or molecular target available? If so, please include it in your discussion.

Authors' Comment

P.5, line 10–14 has been revised and now reads:

“Presented herein, are the synthesis and characterisation of a series of *N*-(4'-nitrophenyl)-*L*-prolinamides via a simple two-step reaction, starting from easily available and inexpensive *p*-halogenonitrobenzenes and *L*-proline. Impelled by Literature precedents^{7,14–18} of prolinamide-containing anti-tumour agents; dating back to actinomycin,¹⁹ four of the most commonly occurring types of human cancers²⁰ (colon, liver, lung and gastric) were also selected for *in vitro* assays. Consequently, the anticancer properties of the synthesised *L*-prolinamides were evaluated against human colon (HCT-116), liver (HepG2), lung (A549) and gastric (SGC7901) carcinoma cell lines.”

Reviewer 1's Comment

3. The activity observed ranges from modest to good, but the potential selectivity could be equally important or even more relevant. Please state whether any selectivity study using cell line models for "healthy cells" was conducted, or state whether that would be part of a planned follow-up study.

Authors' Comment

Selectivity and safety studies on normal cells are slated for future work. The phase reported in the manuscript was aimed at assessing the anticancer properties of the synthesised compounds.

Reviewer 1's Comment

4. For analog 4m, indicate in the structure the stereochemistry of the hydroxyl group at position 4 of the proline.

Authors' Comment

Amended appropriately:

Reviewer 2's Comment

No information on the purity of the compounds is provided. The compounds need HPLC or elemental analysis determined purity levels to ensure that an impurity is not responsible for the assigned activity.

Authors' Comment

The compounds were subjected to column chromatography pre-analyses and their melting point, spectroscopic (¹H- & ¹³C-NMR) and high-resolution mass spectrometric data affirm their purity. The authors are of the conviction that the observed activities (or lack thereof) are not due to impurities.

Reviewer 2's Comment

P4L16 - 'peptides are typically favored over small organic molecules in drug development' This is not true (at least in the way the authors are describing) peptide or peptidomimetic compounds represent a very small % of approved drugs. Small molecules are preferred for the reasons discussed by the authors.

Authors' Comment

Amended. P4L16 now reads:

"Peptides are typically accessed in the development of new classes of therapeutic agents because of their high activity, low immunogenicity, good biocompatibility and amenability..."

Reviewer 2's Comment

The authors screen their compounds at 1, 10 and 100 uM concentrations but why not add another concentration and determine an IC50 value? An IC50 is much more robust and allows better comparison between derivatives.

Authors' Comment

The authors agree with the Reviewer's comment. However, IC50 determination is slated for future work when lead compounds from this phase of the project will also be assayed against normal (non-tumour) cells. In the report, under review, the authors were content to check/compare the activities of the synthesised compounds at low, medium and high concentrations.

Reviewer 2's Comment

The discussion of SAR wherein longer chains provide more active compounds is a common observation in medicinal chemistry. The greater lipophilicity afforded by the longer chains enable greater cell uptake. See Wang et. al. ChemMedChem. 2017, 12, 1033. Consider adding this reference.

Authors' Comment

Suggestion accepted with thanks. P11L48–49 now reads:

"... It was also observed that the longer the *N*'-alkyl chain substituents of the prolinamides (**4a** vs. **4o**), the stronger the antiproliferative activity; in concurrence with previous reports, which suggest that the greater lipophilicity afforded by the longer chains enable greater cell uptake.³⁰ ..."

Reviewer 2's Comment

No selectivity data in a non-cancerous cell line is presented, despite the introduction discussing non-selective toxicity as a real problem. This should be determined at least for the most active compounds. This is especially important with compound 4u which is toxic against all the cells tested. Is this just a toxic compound?

Authors' Comment

The Reviewer's suggestion is noted with thanks. Requisite selectivity/toxicity assays on analogous non-cancerous cell lines will be carried out against active compounds, in the next phase of the project. The phase reported here was aimed at synthesising the compounds and checking for anticancer activity, with a view to finding lead compounds.

Reviewer 2's Comment

P12L21. '4k and 4m suspected of encouraging proliferation' This seems unlikely, is this not just general growth? A negative control in the experiment would determine levels of growth in vehicle alone.

Authors' Comment

A negative control with vehicle alone was included in the experiment and its corresponding cell viability was set to 100. Nonetheless, it is a "point of discussion". The authors intend to carry out further experiments, such as assaying at other varying dilutions.

Reviewer 2's Comment

P12L30; Sulphonamides alone are not a pharmacophore, it is their substituents so this sentence is incorrect.

Authors' Comment

The authors respectfully disagree.

Is it the Reviewer's suggestion that in *N,N*-dimethylbenzenesulphonamide (below), for example, the pharmacophores are the methyl and phenyl substituents?

Reviewer 2's Comment

P12:35; 4l has no inhibition activity or it just wasn't determined as shown in Table 2? It is unlikely that the physical state will be relevant to activity.

Authors' Comment

P12:33–39 has been deleted. P12:39–43 now reads:

“... The % cell viability of the nitrile, *N'*-(2''-cyanophenyl)-*N*-(4'-nitrophenyl)-L-prolinamide **4r** against the four carcinoma cell lines tested ranged from 38.20 ± 6.03% (against HepG2) to 99.20 ± 3.57% (against HCT-116), as shown in Table 2.”

Reviewer 2's Comment

The abstract is very long and switches between discussing cell inhibition % and cell survival % which are not the same thing.

Authors' Comment

The abstract has been abridged to 200 words and now reads:

"Prolinamides are present in secondary metabolites and have wide-ranging biological properties as well as antimicrobial and cytotoxic activities.

N-(4'-substituted phenyl)-*L*-prolinamides **4a–4w** were synthesised in two steps, starting from the condensation of *p*-fluoronitrobenzene **1a–1b** with *L*-proline **2a–2b**, under aqueous–alcoholic basic conditions to afford *N*-aryl-*L*-prolines **3a–3c**, which underwent amidation via a two-stage, one-pot reaction involving SOCl₂ and amines, to furnish *L*-prolinamides in 20–80% yield.

The cytotoxicities of **4a–4w** against four human carcinoma cell lines (SGC7901, HCT-116, HepG2 and A549) were evaluated by MTT assay; with good tumour inhibitory activities (79.50 ± 1.24%–50.04 ± 1.45%) against HepG2. **4a** exhibited the best antitumour activity against A549 with % cell inhibition of 95.41 ± 0.67% at 100 μM. Likewise, **4s** (70.13 ± 3.41%) and **4u** (83.36 ± 1.70%) displayed stronger antineoplastic potencies against A549 than the standard, 5-fluorouracil (64.29 ± 2.09%) whereas **4a** (93.33 ± 1.36%) and **4u** (81.29 ± 2.32%) outperformed the reference (81.20 ± 0.08%) against HCT-116. SGC7901 showed lower % cell viabilities with **4u** (8.02 ± 1.54%) and **4w** (27.27 ± 2.38%). These results underscore the antiproliferative efficacies of *L*-prolinamides whilst exposing **4a** and **4u** as promising broad-spectrum anticancer agents. SAR studies are discussed."

Reviewer 2's Comment

The prolinamide structure in the nucleoside compounds in figure 1 is a little misleading as these are ligands of a protide structure that will be lost in the cell, it has no contribution to the pharmacophore.

Authors' Comment

P4:51–52 has been revised and now reads:

"Figure 1: Some prolinamide-containing anticancer compounds (Abarelix,⁷ Phosmidosine¹⁰ and Pro-4¹⁴)"

Reviewer 2's Comment

Scheme 1 shows 3a-d but the text only refers to 3a-c.

Authors' Comment

Scheme 1 now reads:

Reviewer 2's Comment

A line to separate cell types in table 2 would be useful for clarity of presentation.

Authors' Comment

Done.